# Nuclear quantum effects on zeolite proton hopping kinetics explored with machine learning potentials and path integral molecular dynamics

Massimo Bocus [1,2], Ruben Goeminne[1,2], Aran Lamaire [1],
Maarten Cools-Ceuppens[1], Toon Verstraelen[1] & Veronique Van Speybroeck [1] ✉

Proton hopping is a key reactive process within zeolite catalysis. However, the accurate determination of its kinetics poses major challenges both for theoreticians and experimentalists. Nuclear quantum effects (NQEs) are known to influence the structure and dynamics of protons, but their rigorous inclusion through the path integral molecular dynamics (PIMD) formalism was so far beyond reach for zeolite catalyzed processes due to the excessive computational cost of evaluating all forces and energies at the Density Functional Theory (DFT) level. Herein, we overcome this limitation by training first a reactive machine learning potential (MLP) that can reproduce with high fidelity the DFT potential energy surface of proton hopping around the first Al coordination sphere in the H-CHA zeolite. The MLP offers an immense computational speedup, enabling us to derive accurate reaction kinetics beyond standard transition state theory for the proton hopping reaction. Overall, more than 0.6 μs of simulation time was needed, which is far beyond reach of any standard DFT approach. NQEs are found to significantly impact the proton hopping kinetics up to ~473 K. Moreover, PIMD simulations with deuterium can be performed without any additional training to compute kinetic isotope effects over a broad range of temperatures.

Brønsted-acidic zeolites are versatile, resistant catalysts that for decades have been recognized as the workhorse of the petrochemical industry[1]. Furthermore, they are also expected to play a vital role in next-generation biorefineries for the conversion of non-fossil feedstocks[2]. From a theoretical point of view, zeolites belong to the most studied materials in the field of heterogeneous computational catalysis[3]. In their ideal, defect-free crystalline form, Brønsted-acidic zeolites are composed of interconnected $SiO_4$ tetrahedra, where a fraction of the $Si^{4+}$ ions is substituted by $Al^{3+}$. The excess of negative charge is compensated by the addition of a proton—the Brønsted Acid Site (BAS)—on one of the oxygens in the first coordination sphere of the Al substitution. Interestingly, the BAS is not confined to a specific oxygen of the Al tetrahedron, but it can jump from one oxygen atom to another in what is commonly known as the 'proton hopping' reaction. This process is one of the most fundamental activated events within zeolite chemistry (Fig. 1a) and represents the archetypal proton-transfer reaction which is at the base of any Brønsted acid-catalyzed reaction.

Because of its apparent simplicity, proton hopping is an ideal case study for both experiment and theory, hence various methods have been used to investigate the process kinetics. Experimentally, Nuclear Magnetic Resonance (NMR)[4–6], Impedance Spectroscopy (IS)[7] and

[1]Center for Molecular Modeling, Ghent University, Technologiepark 46, 9052 Zwijnaarde, Belgium. [2]These authors contributed equally: Massimo Bocus, Ruben Goeminne. ✉e-mail: Veronique.Vanspeybroeck@Ugent.be

**Fig. 1 | Poor agreement is found in the available literature for the activation energy of the proton hopping reaction. a** Schematic depiction of the proton hopping reaction. **b** Activation energy for the proton hopping process as function of the Si/Al ratio for multiple zeolites, as derived from the available literature[75]. The data was obtained from ab initio calculations (blue, the points with an asterisk correspond to cluster calculations and, therefore, the Si/Al ratio is meaningless) or IR (red), NMR (yellow) and IS (green) spectroscopies. If more values are available for different temperatures, they are reported with diamonds interconnected by a dotted line. For more details about the reported values and a full list of references, the interested reader is referred to Supplementary Note 1. **c** Part of the H-CHA unit cell, showing the conventional nomenclature of the oxygen atoms in the first coordination sphere of the Al defect adopted herein (Si is in yellow, O in red, Al in blue and H in white).

InfraRed spectroscopy (IR)[8] have been employed to retrieve the activation energies for the proton hopping process. From the theoretical side, the reaction has been tackled with various methodologies ranging from static simulations[9–15] to enhanced-sampling techniques based on molecular dynamics (MD)[16]. Given this plethora of scientific reports, it would be tempting to assume that every detail of the proton hopping reaction is now revealed. However, when surveying the available literature, a huge spread in both the theoretically and experimentally obtained activation energies for proton hopping barriers can be found (Fig. 1b).

In general, activation energies derived from NMR spectroscopy are lower than the theoretical ones. From IR experiments, two different activation energies were obtained for two different temperature ranges (398–548 and 573–773 K, see red diamonds in Fig. 1), a fact that was attributed to the switch from intra-site hopping to inter-site hopping[8]. However, a more recent investigation has disproven such interpretation and indicated active site proximity effects as the main cause for the observed change in activation energy[17]. Inter-site hopping was also suggested to be responsible for the high activation energies retrieved with IS[7].

To understand this lack of consistency, it is important to consider the main possible sources of discrepancy between the proton hopping barriers from literature. First, the residual presence of water in the catalyst is often indicated as the main source for the—in general—low experimental barriers[10], as it is almost impossible to achieve a completely dry material with routine drying procedures[18]. Moreover, the presence of defective sites like extra-framework aluminum species is known to alter the BAS' acidity compared to the pristine material[19]. On the theoretical side, most of the calculations performed so far did not explicitly account for the quantum nature of the hydrogen nucleus. Instead, the nuclei in the system are treated as classical particles moving on an underlying Potential Energy Surface (PES), which is obtained by solving the electronic many-body problem using quantum many-body techniques. This is normally done using Density Functional Theory (DFT) for the sake of computational efficiency. In what follows, the terminology 'classical DFT PES' will be used to refer to nuclei that are treated as classical particles on a DFT-determined PES, thus the electronic degrees of freedom are treated quantum mechanically whereas the nuclei are treated as classical particles. To include Nuclear Quantum Effects (NQEs), approximative methods have been used. For

example, tunneling corrections have sometimes been applied to account for NQEs[12]. To more rigorously account for NQEs, one should resort to computationally more expensive methods such as the Path Integral Molecular Dynamics (PIMD) approach, which relies on Feynman's path integral formulation of quantum mechanics. Within PIMD simulations, the statistics of quantum particles are retrieved using a classical ring polymer consisting of $P$ replicas of the system[20]. Each replica runs on the classical DFT PES, making PIMD $P$ times more expensive than a standard MD simulation. This is because an independent DFT-level energy and force evaluation must be performed every MD step for each replica. Within the field of heterogeneous-catalyzed reactions such simulations have so far been mostly unfeasible due to the high computational cost of each PIMD step, as at least 10 replicas are usually required to achieve converged results—making the simulations prohibitively expensive[21]. Nonetheless, there is clear evidence that NQEs may have a significant impact on the physico-chemical properties of systems containing light atoms[20,22–24]. For example, it is well-known that they can significantly affect the strength of hydrogen bonds in a variety of systems[25,26]. NQEs have never been explicitly included in zeolite-related reactions and thus it remains unclear to what extent they would affect the rate of proton hopping and by extension any proton-transfer reactions within the field of zeolite catalysis.

To fill this gap of knowledge, we present in the current contribution a methodology that may allow to systematically include NQEs when investigating activated hydrogen transfer events. To this end, proton hopping in H-CHA with isolated active sites is used as a case study (Fig. 1c), for which we first trained an accurate Machine Learning Potential (MLP) based on an underlying set of high-temperature DFT Umbrella Sampling (US) simulations. The use of enhanced-sampling simulations is essential to explore in an efficient way the less-probable highly energetic regions of the PES, which are typically associated with reactive events. The underlying DFT simulations at finite temperatures serve as input to train a deep neural network MLP (Fig. 2). Once an accurate MLP is constructed, an enormous computational speedup can be achieved, which allowed to: (i) compute the Free Energy Surfaces (FESs) of all possible hoppings around an isolated Al defect in the temperature range 273–873 K with a large number of umbrellas and long simulation times to obtain well-converged results, (ii) explicitly include NQEs through the PIMD approach, (iii) derive accurate kinetic

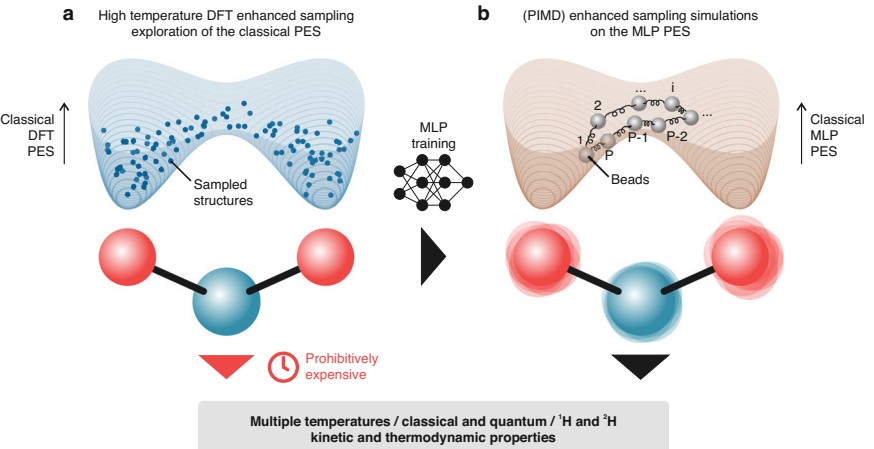

**a** High temperature DFT enhanced sampling exploration of the classical PES

**b** (PIMD) enhanced sampling simulations on the MLP PES

**Fig. 2 | Kinetic and thermodynamic properties with and without NQEs over a wide range of temperatures can be computed using an MLP trained on high-temperature enhanced-sampling DFT simulations.** The pictures show a simplified schematic representation of the PES experienced by the proton when hopping between two oxygens around the Al site (red and blue spheres, respectively), sampled with DFT simulations (**a**) and subsequently learned with the MLP (**b**). In **b** a schematic view of the hydrogen ring polymer with P beads running on the classical MLP PES is shown.

constants beyond the Transition State Theory (TST) approximation, taking barrier recrossing into account via the reactive flux formalism[27] and (iv) perform an additional set of simulations with deuterium instead of protium to explicitly compute the Kinetic Isotope Effect (KIE) on the reaction.

We show that even at catalytically relevant temperatures (>400 K) NQEs may still be important to consider when computing reaction kinetics and their relevance is not restricted to the absolute low temperature regime. While the work performed here is illustrative for the most basic proton hopping reaction in zeolites, it provides the means to routinely include NQEs and explicitly calculate KIEs when studying any proton-transfer event in heterogeneous catalysis.

## Results

### Construction of a reactive MLP with DFT accuracy

To train an accurate MLP, a sufficiently large set of DFT datapoints is required, which should cover the relevant regions of the reaction phase space. To this end, high-temperature (873 K) DFT US simulations were performed on a CHA conventional cell containing 36 T Si atoms, where 1 silicon is replaced by Al to give a final Si/Al ratio of 35. The temperature choice of 873 K is arbitrary but, in general, on the higher end of typical zeolite-catalyzed processes[1]. In the CHA topology, all T atoms are equivalent. However, the four O atoms in the first coordination sphere of the Al defect are not (Fig. 1c). This leads to 6 distinct hopping paths, which are all considered in this work. To assess whether any path could be significantly disfavored, activation free energies were initially screened with static calculations. The results suggest that all 6 possible hopping paths have relatively similar activation free energies (within ~20 kJ mol$^{-1}$) and no single one is strongly (dis)favored (Supplementary Note 2). Therefore, 6 separate DFT US simulations at 873 K were performed to sample all the possible hoppings. A difference in coordination numbers between the proton and the two oxygen atoms involved in the hopping was used as main collective variable to bias the system (see "Methods" section and Supplementary Note 3). One-dimensional umbrellas were used to sample the reaction path and, if needed, additional two-dimensional umbrellas were added to improve the sampling of scarcely visited regions of the phase space (more details are reported in Supplementary Note 3.2). A full overview of the DFT US results is reported in Supplementary Note 4.

Energies and forces were extracted every 5 fs from the DFT US trajectories, yielding a total of ~1,200,000 structures which were used to train an MLP with the SchNetPack package (see "Methods" section and Fig. 2)[28,29]. Performing MD simulations with the MLP provides a dramatic speedup in computational time, going from ~8.3 s/step on 56 Xeon E5-2680v4 CPUs@2.4 GHz cores to ~0.01 s/step on a single NVIDIA Volta V100 GPU. As part of the MLP validation, well-converged 873 K DFT FESs were generated to compare them with the MLP-derived ones within a reasonably small uncertainty. To this end, about 50 ps of simulation time was required for each DFT umbrella. Considering that 19 umbrellas are needed to sample each of the 6 hoppings, this was a computationally demanding task. On the other hand, it also provided us with a very large number of DFT datapoints, hence the large number of structures used to train the MLP. With the acquired knowledge that a mean absolute error on the force of about 40 meV Å$^{-1}$ is sufficient to obtain very accurate FESs (vide infra), we also tested the performance of newer and more data efficient equivariant neural networks[30], where preliminary results indicate that a few hundred fs per umbrella are sufficient to achieve converged results (Supplementary Note 10), providing an enormous computational saving in the DFT data generation.

To further validate the trained MLP, we also tested whether it could reproduce FESs at lower temperatures than the training one. To this end, three additional sets of DFT US simulations were performed. The 2–3 hopping was tested at 573 and 273 K, while the 1–4 hopping was tested at 273 K. With this choice, the hoppings with the smallest activation energies are tested and all four oxygens are considered at the lowest temperature. For the sake of clarity, a detailed comparison between the MLP and DFT results is presented in Supplementary Note 6.1, while here only the 2–3 hopping is discussed in detail. As shown in Fig. 3, the DFT and MLP FESs exhibit an almost perfect overlap, with most variations contained well within the error bars. The free energy barrier exhibits a clear increase with temperature, which is in line with a rigid transition state associated with a negative entropy variation. It must be pointed out that, thanks to the large computational speedup enabled by the MLP, longer simulation times (100 ps vs. 50 ps per umbrella) and a larger number of umbrellas (39 vs. 19 per hopping) were easily achievable. This led to a vastly improved sampling of the reaction PES, thereby obtaining much better converged FESs. Moreover, all MLP simulations were repeated three separate times starting from different initial velocities and the associated results and uncertainties were obtained by averaging over these three independent runs. Initially, the 573 K DFT profile presented a moderate spike in the transition state region, which was not present in the MLP profile. Therefore, 2 additional umbrellas were added in the proximity of the transition state and an additional 40 ps of simulation was performed in every umbrella, for a total of 90 ps. The final DFT profile

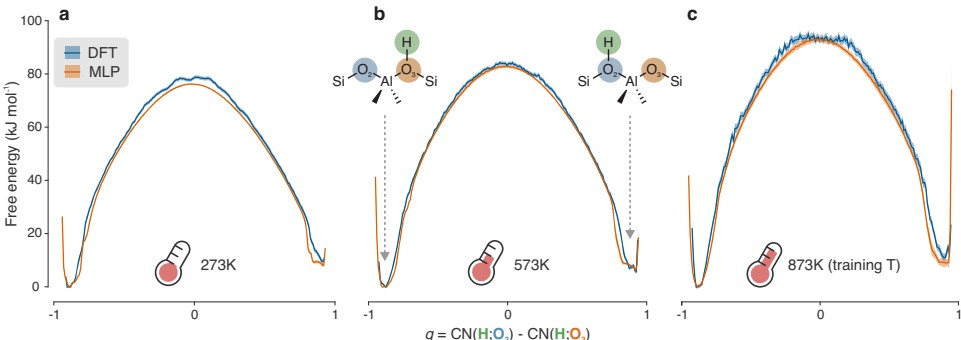

**Fig. 3 | The trained MLP can reproduce DFT FESs with high accuracy.** The MLP and DFT FESs (in orange and blue, respectively) of the 2–3 hopping at 273 (**a**), 573 (**b**) and 873 K (**c**) are almost perfectly superimposable, even though the MLP model is only trained on snapshots at the highest temperature. A comparison with the other available FESs is shown in Supplementary Fig. 10. For a more detailed definition of the collective variable (*q*) used to build the FESs, the interested reader is referred to the "Methods" section. Uncertainties (shaded areas around the FESs) on the MLP FESs are obtained by averaging over three independent runs (Supplementary Note 6.1) while for DFT they are estimated according to the procedure summarized in Supplementary Note 3.1.3. Source data are provided as Source Data File.

reaches almost perfect agreement with the MLP one, highlighting how the (small) differences between MLP and DFT FESs are almost certainly caused by sampling issues rather than by significant deviations in the underlying PES. The results show that (i) the trained MLP is effectively capable of encoding chemical reactivity and (ii) high accuracy on the computed FESs is retained also for temperatures lower than the training one, offering thermodynamic transferability in terms of operating conditions.

While directly superimposing FESs provides an intuitive visual means of comparison, the FES itself is not experimentally measurable. The final macroscopic quantity of interest is the kinetic constant of the reaction, which does not depend on the choice of the collective variable used to represent the FES[31,32]. By means of classical (TST), the forward and backward kinetic constants for the 6 high-temperature hoppings, the 2–3 hopping at 573 K and the 1–4 and 2–3 hoppings at 273 K were retrieved (see "Methods" section). Fig 4 reports a graphical comparison between the DFT and MLP rates, where the computed kinetic constants are converted to a corresponding phenomenological barrier using Eyring's equation[32]. The sole purpose of the latter is to provide a more tangible equivalent to the kinetic constant, without comparing values that can span multiple orders of magnitude (more details are provided in the "Methods" section). None of the computed barriers differ more than ~5 kJ mol⁻¹ and, for most of the hoppings, the MLP values lie within the error bars of the DFT ones. These results indicate that the MLP accurately reproduces the DFT PES underlying the proton hopping reaction in H-CHA and can therefore be used to compute reaction rates at any temperature of interest and to explicitly introduce NQEs through the PIMD approach (Fig. 2).

### Full characterization of the hopping kinetics

Having validated the MLP to faithfully reproduce the proton hopping FESs over a broad temperature range (273-873 K), additional US simulations were performed to retrieve the full reaction kinetics considering all hopping paths. Moreover, NQEs can be systematically included in the reaction investigation as the PIMD formalism becomes accessible, thanks to the large computational efficiency of the MLP. To obtain well-converged FESs, at least 16 system replicas (also referred to as beads) are required in the ring polymer (Supplementary Note 7.1). A graphical visualization of the spread of the beads around the transition state region compared with the classical case is shown in Fig. 5, where it becomes clear that quite some uncertainty is present on the proton's position compared to the classical deterministic trajectories. A full overview of the classical and quantum FESs is reported in Supplementary Notes 6.1 and 7.3, respectively. Introducing NQEs leads to a general decrease of the free energy barriers compared to the case

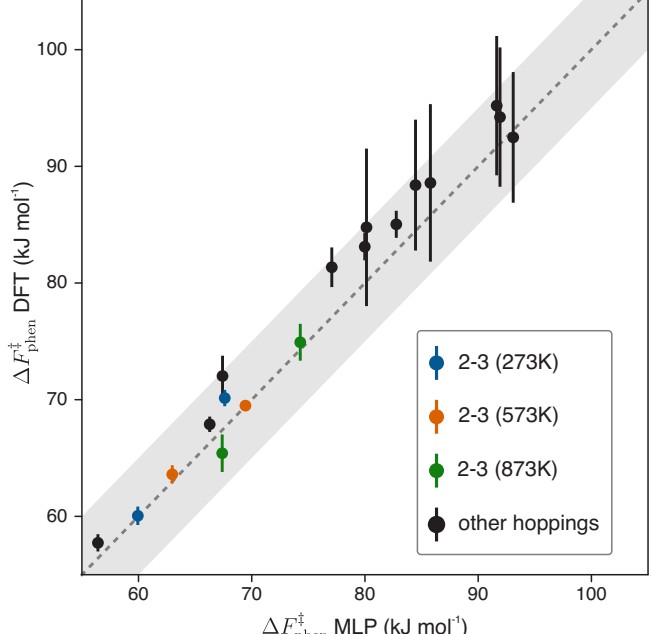

**Fig. 4 | The DFT and MLP phenomenological barriers are in excellent agreement.** The available DFT phenomenological barriers (12 forward and backward barriers for the 873 K hoppings, the forward and backward barriers of the 2–3 hopping at 573 K and the forward and backward barrier of the 1–4 and 2–3 hoppings at 273 K are all within about ±5 kJ mol⁻¹ from the MLP ones, as highlighted by the gray shaded area. The phenomenological barriers of the 2–3 hopping at various temperatures, corresponding with the free energy profiles of Fig. 3, are highlighted in different colors (273 K in blue, 573 K in orange and 873 K in green). Uncertainties on the MLP barriers are obtained by averaging over three independent runs (although their magnitude is so low that they are barely visible in the figure) while for DFT they are estimated according to the procedure summarized in Supplementary Note 3.3. Source data are provided as Source Data File.

where nuclei are treated classically. This effect tends to lessen with increasing temperatures, in accordance with the expected convergence between the quantum and classical behavior for high temperatures.

Not only does the MLP allow to include in a reliable—yet computationally feasible—way NQEs, but it also allows to determine reaction kinetics beyond classical TST and explicitly include barrier recrossing through the reactive flux formalism (obtaining the true

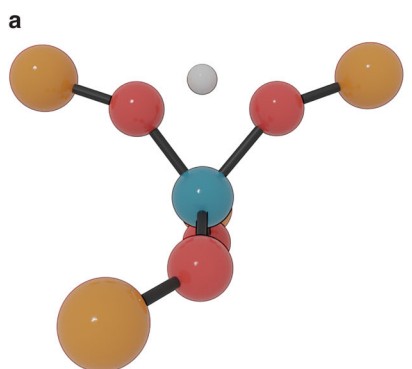

**Fig. 5 | The use of PIMD leads to a significant spread in the proton location.**
These two snapshots, arbitrarily extracted from the transition state umbrella of the
2−3 hopping, highlight how the system beads in PIMD (**b**) can be spread quite
significantly in space with respect to the classical case (**a**). For the sake of clarity,
only the H-SSZ-13 atoms up to the second coordination sphere around the Al site
are shown and in **b** a superposition of all beads is only present for the proton. Si is
depicted in yellow, O in red, Al in blue and H in white.

kinetic constant of the reaction)[27,33], which is in its turn part of the
Bennett-Chandler reaction rate theory[34,35]. In this approach, multi-
ple unbiased simulations (5000 in this case) are initialized atop the
transition state and monitored through time, to retrieve whether
they end up in the product basin or whether they recross the barrier
towards the reactant basin (see "Methods" section). This approach
is most appropriate when NQEs are included, as quantum TST
approximations such as ring polymer molecular dynamics (RPMD)
TST do not yield a strict upper bound for the quantum rate (more
details can be found in Supplementary Note 7.4)[36]. Overall, the MLP
US simulations allowed to compute three different kinetic con-
stants for all hopping paths and all temperatures: a classical TST-
based one ($k_{TST}^c$), derived from classical MD and the TST approx-
imation, a classical Bennett-Chandler one ($k_{BC}^c$), where barrier
recrossing is now explicitly taken into account and, finally, a
quantum Bennett-Chandler one ($k_{BC}^q$), analogous to $k_{BC}^c$ but derived
from the RPMD simulations and thus including NQEs. Remark that
the amount of data used to obtain them is well beyond the reach of
any pure ab initio methodology where all energy and force evalua-
tions are performed at the DFT level. Even when excluding the
thousands of short trajectories required to obtain well-converged
$k_{BC}$ values, computing the quantum FESs requires 42 sets of US
simulations (6 hopping paths at 7 different temperatures), each
consisting of 39 umbrellas simulated with 16 parallel beads−for a
total of more than 0.6 μs of simulation time. Such simulation
lengths are clearly beyond the limit of any DFT-based MD
simulation.

While demonstrating the impact of NQEs on the reaction rate is
important to highlight the cases in which NQEs cannot be neglected
and should thus be accounted for computationally, the resulting
'quantum speedup' is not experimentally measurable as NQEs are an
intrinsic part of nature. What is often measured experimentally, on
the other hand, is the KIE−namely the change in rate when the
hydrogen atoms in the system are substituted with deuterium
(other isotopic substitutions are of course also possible[37]). Inter-
estingly, the MLP trained on [1]H simulations can be directly used for
KIE calculations, as the reaction PES does not depend on the atomic
masses but only on their charge. An additional set of PIMD simula-
tions was therefore performed at 273, 573 and 873 K with the BAS
mass set to 2 a.m.u. Given the linear behavior of $\ln(k_{BC}^q)$ over the
whole temperature range (vide infra, Fig. 7), the intermediate
temperatures were no longer considered. A full overview of
the simulations' results is reported in Supplementary Note 7.5. The
error on the MLP forces with respect to DFT remains basically
unaffected by the change in the hydrogen mass, confirming that
both simulations sample analogous PES regions (Supplementary

Fig. 20). The reactive flux kinetic constant for the PIMD simulations
with deuterium will be indicated with $k_{BC}^q(^2H)$, while for protium
simulations the isotope label will be omitted.

Using the computed kinetic constants, the equilibrium coverage
of the 4 oxygen sites ($\theta_i$, $i = 1 − 4$) was determined as a function of
temperature (more details can be found in Supplementary Note 8).
The results for [1]H are shown in Fig. 6a−c. When considering the $k_{TST}^c$
and $k_{BC}^c$ kinetic constants, similar equilibrium populations are
obtained within the limits of uncertainty (Fig. 6a, b), which is a con-
sequence of the similar recrossing rate between the forward and
backward barriers. In general, $O_1$ and $O_3$ are the most populated sites at
any temperature, followed by $O_2$ and $O_4$. In the classical case, $O_3$ has
the largest population up to 373 K, while at higher temperatures its
population becomes nearly identical to $O_1$. Significantly different
results are obtained when the quantum kinetic constants ($k_{BC}^q$, Fig. 6c)
are considered, where $\theta_3$ remains significantly larger than $\theta_1$ at all
temperatures. A similar trend is obtained for the PIMD simulations
with deuterium (Supplementary Fig. 19). When the proton is on $O_3$, it
finds itself oriented towards the center of the 6 T atoms ring (Fig. 6d)
and can, therefore, interact with the oxygens on the opposite side. To
understand more profoundly the impact of these intra-framework
interactions, we performed a 273 K classical MD and PIMD simulations
of the zeolite with the proton located on $O_3$. We then analyzed the
radial distribution functions (RDFs) of the proton with the 6 oxygens
sharing the Si and Al with $O_3$ ('adjacent' in Fig. 6e) and all the other
oxygens in the unit cell ('others'). We found that−as expected− the
BAS lies within 2−3 Å from the oxygens on the other side of the 6-Si ring
and will therefore interact with them. By comparing the classical and
quantum RDFs (Fig. 6e), it can be seen how the maximum in the RDF
H-O (others) occurs at slightly shorter distances in the quantum case
and, moreover, shorter distances−in the order of ~2 Å−are explored
more often. Based on these findings, it appears that when the quantum
nature of the hydrogen nucleus is considered the weak interaction
between the non-adjacent framework oxygens and the proton
becomes stronger and, as a result, $O_3$ becomes further stabilized with
an increase in $\theta_3$. No other site interacts with other framework oxygens
within 3 Å (Supplementary Fig. 18). Previous reports in the literature,
based on geometrical considerations concerning the crystallographic
zeolite unit cell, suggested that none of the four BAS locations are
suited to form H-bonds with other oxygens in the framework[38]. This no
longer seems to be the case when temperature effects and NQEs are
explicitly taken into account. The number of zeolite frameworks pre-
senting this type of intra-framework interaction could thus be higher
than previously thought[38] at realistic operando conditions.

Once all the equilibrium coverages as a function of temperature
are known, the overall hopping rate can be computed using the

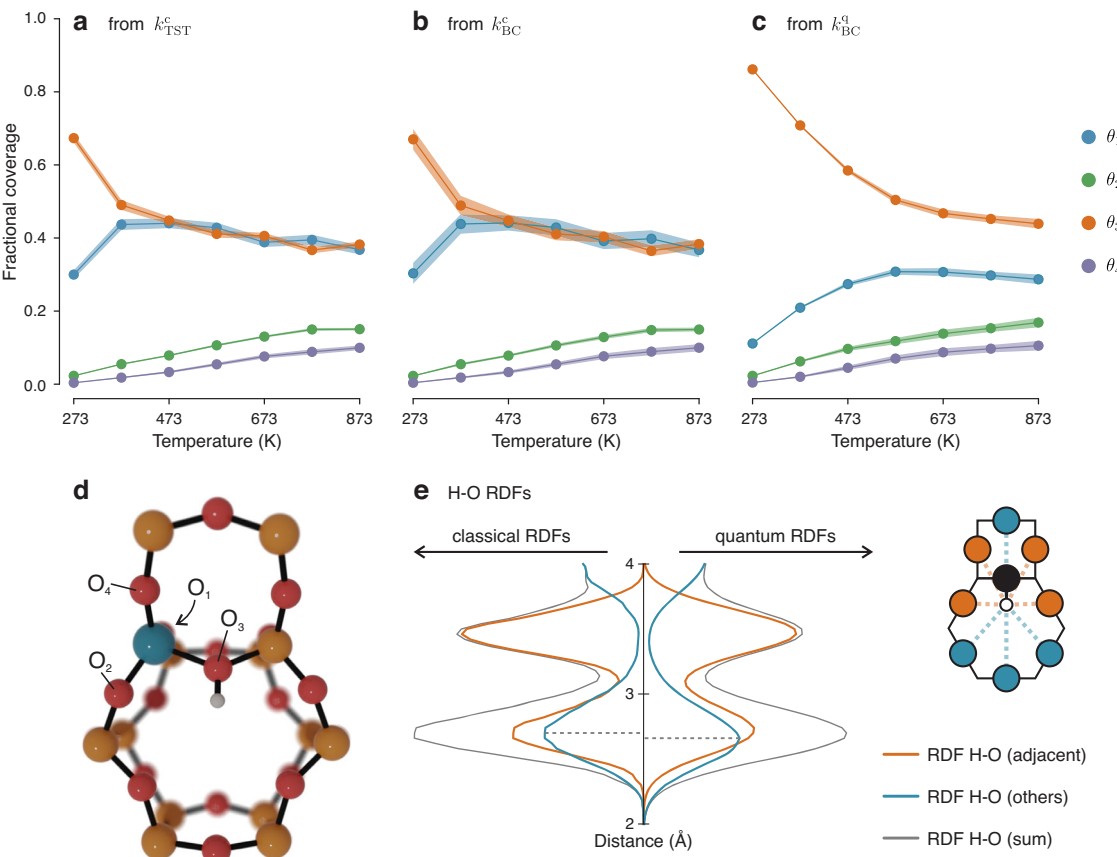

**Fig. 6 | Including NQEs can change the computed equilibrium proton coverages. a–c** Equilibrium coverages ($\theta_i$) of the 4 oxygen sites around the Al defect as function of temperature, computed from the $k_{TST}^c$, $k_{BC}^c$ and $k_{BC}^q$ kinetic constants. **d** Portion of the H-CHA framework as seen along the c cell vector, showing the optimized structure of the BAS on $O_3$. Si is depicted in yellow, O in red, Al in blue and H in white. **e** RDFs between the BAS lying on $O_3$ and the 6 oxygens sharing the same Si and Al as $O_3$ (orange line) and the BAS and all other oxygens in the unit cell (blue line), from classical and NQE simulations. The schematic drawing is seen from the same perspective as d and highlights some of the oxygens belonging to 'adjacent' and 'others'. Source data are provided as Source Data File.

formula:

$$r = \sum_{i=1}^{4} \sum_{j \neq i}^{4} k_{ij} \theta_i, \qquad (1)$$

where $k_{ij}$ is the kinetic constant of the hopping from $O_i$ to $O_j$ and $\theta_i$ the coverage of $O_i$. From this, an Arrhenius plot for the hopping rate as a function of the temperature is computed (Fig. 7a), whose activation energy should be comparable with experiment. First, we analyzed in how far each of the hoppings is contributing to the overall rate. In all cases, only two hopping paths dominate the rate kinetics (Fig. 7b), namely the 1↔4 and 2↔3 paths, as one could expect based on their low free energy barriers. Note that the forward and backward rates have similar contributions, as a higher free energy of the minimum corresponds to both a lower coverage and a lower free energy barrier to hop towards a stabler minimum. These two factors tend to cancel each other when computing $k_{ij}\theta_i$. Minor contributions are given by the 1↔2 and 2↔4 paths, while the remaining two paths (1↔3 and 3↔4) only have noticeable contributions at the highest temperatures. In the deuterium case, the 2↔3 path becomes even more dominant at the expenses of 1↔4 as it appears that the transition state energy is not shifted consistently by the isotope substitution (Supplementary Fig. 18).

By considering the slope of the best fit lines in the Arrhenius plots (Fig. 7a) it is possible to retrieve an effective activation energy for the proton hopping reaction. The $k_{TST}^c$ results yield an activation energy of 67.1 kJ mol⁻¹. Going beyond the TST approximation and explicitly

including recrossing ($k_{BC}^c$) does not significantly change the results, with a consistent—but almost negligible—decrease in the rate across the whole temperature range. When NQEs are included ($k_{BC}^q$), in contrast, the activation energy decreases with about 11 kJ mol⁻¹ due to the possibility of the proton to tunnel through the potential energy barriers. When analyzing the rates related to a specific hopping (Supplementary Fig. 17), it was noticed that this effect is not constant, and becomes more prominent in the hopping paths with a more sharply peaked FES around the transition state region (Supplementary Figs. 15 and 16). This is because a narrower barrier increases the probability of tunneling, which in practice means that the beads of the ring polymer are easily located on both sides of the potential energy barrier experiencing on average a lower free energy. These results show that the impact of NQEs is not systematic in nature and can therefore be challenging to capture with ad hoc corrections. Indeed, previous investigations which included NQEs through an a posteriori tunneling correction suggested that above room temperature no significant effect should be observed[12]. Our results, on the other hand, show that the reaction proceeds 65 times faster at 273 K if NQEs are included and, even at 373 K, a 16-fold increase in the computed rate is still present (Fig. 7c). At 473 K, the reaction remains 7 times faster while the speedup, as expected, tends to become negligible at higher temperatures. It appears therefore that for zeolite-catalyzed processes conducted at milder conditions, among which the ones related to biomass conversion are a predominant example[2], NQEs might be non-negligible when computing the kinetics of proton-transfer steps. A few examples where these effects might be important are the aqueous cyclohexanol

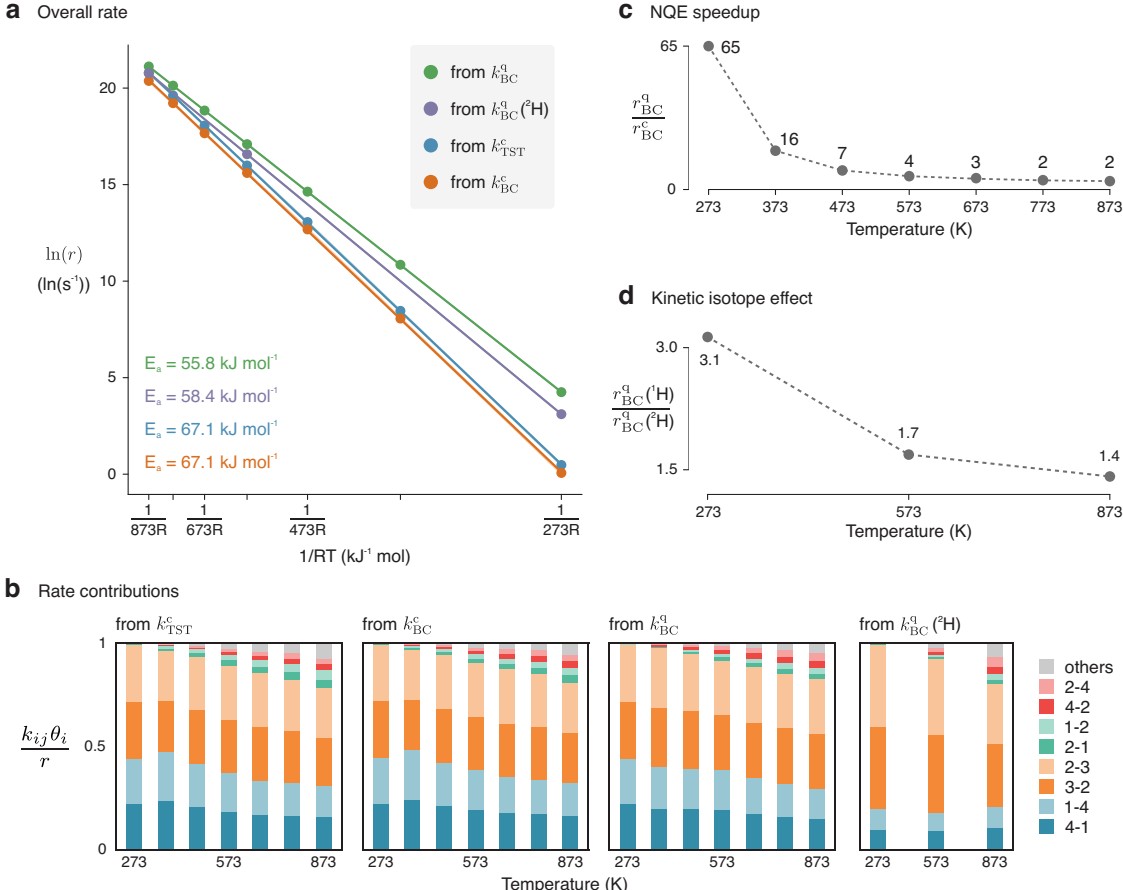

**Fig. 7 | NQEs have a significant effect on the computed hopping rate.**
**a** Arrhenius plot showing the overall hopping rate as function of 1/RT.
**b** Contributions to the overall rate from the different hopping paths, as function of temperature and methodology used. **c** Speedup in the rate due to the inclusion of the NQEs. **d** KIE as function of reaction temperature. Source data are provided as Source Data File.

dehydration in H-Beta zeolite at temperatures of about 400 K[39] and the oxidation of methane to methanol in copper-loaded zeolites carried out at ~473 K[40]. These two cases serve as examples of the relevant application area and conditions influenced by NQEs in zeolites. However, it is clear that many more cases could be affected by the inclusion of NQEs.

When protium is substituted by deuterium ($k_{BC}^{q}(^2H)$) the activation energy becomes 58.4 kJ mol$^{-1}$, yielding a small increase of 2.6 kJ mol$^{-1}$. This is in line with the modest magnitude of the predicted KIE (given by the ratio between the total rate with protium and with deuterium, Fig. 7d), which equals 3.1 at 273 K and decreases to 1.4 at 873 K, in line with standard Bigeleisen-Mayer KIE theory[41,42]. It can be noted how the deuterium rates remain significantly larger than the classical $k_{BC}^{c}$ ones, indicating that the increase in mass is not sufficient to fully suppress the quantum behavior. Unfortunately, there has not been any experimental attempt so far to measure the KIE for proton hopping in zeolites. Theoretically, an early investigation based on static cluster calculations was performed by Fermann and Auerbach[43]. According to their semiclassical TST model, the differences between $^1H$ and $^2H$ are limited above room temperature, in line with our results.

So far, the study focused on H-CHA, which has a single indistinguishable T-site and a small unit cell. To broaden our scope, it is important to assess the MLP capability of describing other zeolite frameworks on which the MLP was not trained. To this end, the transferability of the MLP to other zeolite topologies was investigated. More specifically, we selected five all-silica frameworks from the international zeolite (IZA) database[44] that are of interest for catalytic applications[3] (AFX, CHA, FER, MFI and MOR) and performed a 100 ps

NVT DFT MD simulation using the crystallographic unit cell parameters reported in the IZA database (see Supplementary Note 9). The CHA topology was included as a control system, to ensure that the MLP is robust with respect to changes in the unit cell volume. None of the MLP simulations presented obvious instabilities and the error on the forces is not excessive, even for frameworks that do not share any secondary building unit with CHA, varying between 196 meV Å$^{-1}$ for MOR and 258 meV Å$^{-1}$ for MFI. The quality of the zeolite trajectories, monitored through the Si–O and Si–Si RDFs, remains reasonably good with only small long-range differences for MFI (Supplementary Fig. 19). Testing the proton hopping reactivity in a systematic way for more frameworks would require a further set of expensive ab initio US simulations and, therefore, is outside the scope of this work. The results obtained on the all-silica frameworks, nonetheless, still indicate that the MLP can capture to a large extent the chemistry of Si–O–Si bonds and, therefore, we expect that not many additional DFT simulations would be needed to retrain it and extend its accuracy to new zeolite frameworks, for instance building on the transferable MLP for siliceous frameworks by Erlebach et al.[45] towards aluminum-containing zeolites of catalytic interest.

## Discussion
Proton-transfer reactions are of primordial importance within zeolite catalysis. Thus far, it was unclear in how far NQEs affect the barriers and rates of proton hopping processes at realistic operating conditions, as their explicit inclusion through PIMD was prohibitively expensive if the underlying classical PES is evaluated at a DFT level of theory. Herein, we showed that a reactive MLP can be trained based on

underlying high-temperature (873 K) US simulations at the DFT level, that provides kinetic results with a similar accuracy as the underlying DFT data. However, thanks to the enormous computational speedup gained by describing the PES based on the MLP compared to the original DFT energy and force calculations, the MLP can be used to perform virtually any type of simulation that relies on the classical PES of the considered reaction(s) over a broad range of temperatures. The proposed methodology thus not only succeeds in reproducing the underlying DFT simulations but comes with a series of advantages that were so far unreachable due to the prohibitively excessive computational cost.

First, the convergence of the free energy surfaces obtained from enhanced-sampling techniques can be improved by using many more umbrellas and by simulating for a longer time. Secondly, PIMD can be employed to explicitly account for the quantum nature of the nuclei in the system. While the inclusion of NQEs through MLPs has already been proposed in the literature[22,46,47], the application of PIMD/MLP to an activated event in heterogenous catalysis was still unexplored. We remark that for simulations at cryogenic temperatures the number of beads required to achieve converged results could become very large even for the MLP. This problem can be mitigated by coupling the MLP simulations with path integral acceleration techniques[20]. Thirdly, it also becomes possible to go beyond the TST approximation and explicitly include barrier recrossing via the reactive flux formalism, thereby obtaining the true kinetic constant of the reaction. Because of the thousands of short MD trajectories that have to be initialized atop the transition state, this type of calculation was so far too expensive to be performed at a DFT level of theory. The more efficient methodology for describing the forces and energies may also open the window to use methods like transition path sampling within the field of zeolite catalysis, which were thus far not truly accessible due to the large number of paths that needs to be sampled at the DFT level[48]. Finally, KIEs can be explicitly computed if the PIMD simulations are performed with different nuclear masses, as this does not affect the underlying PES learned by the MLP.

Our results show that the expected Arrhenius activation energy for the hopping process, considering all six hoppings and the coverages of the four oxygen sites, is 67.1 kJ mol$^{-1}$ in the absence of NQEs, whereas including the quantum nature of the proton brings the activation energy down to 55.8 kJ mol$^{-1}$. When quantitively comparing this activation energy to experimental results, it is important to note that this study makes use of the revPBE-D3 level of theory, which is known to underestimate the activation energies of chemical reactions[16,49,50]. In this sense, our barriers will present a lower boundary for the chemically accurate activation energy. Because of the large improvement in data efficiency of newer MLP architectures (Section S10 of the Supplementary Information), we believe that training an accurate model based on a more expensive albeit more reliable exchange-correlation functionals should become feasible. The computed activation energy remains relatively higher than the experimentally available ones. The most likely source of discrepancy lies in the perfect crystalline nature of the adopted zeolite model. The presence of residual water molecules, defects (EFAL species, for instance) and an heterogeneous aluminum distribution are basically unavoidable at the macroscale and all these factors are known to potentially affect the behavior of protons in zeolites[17,19,51]. According to the simulations, a primary KIE of about 3 is expected at 273 K but no experimental evidence is available thus far to corroborate this result.

This proof-of-concept study presents a general scheme to obtain MLP models that can simulate proton hoppings and activated processes in zeolite catalysis with improved realism. The proposed methodology is, in principle, extendible to additional reactions and reactive environments, making it a valuable tool for studying a wide range of catalytic phenomena[52].

## Methods

### Umbrella sampling simulations

The hopping of the H-CHA BAS between the oxygens in the first coordination sphere of the Al defect was studied by means of umbrella sampling simulations[53,54]. In this approach, quadratic bias potentials (the 'umbrellas') are placed along a certain collective variable ($q$) which should smoothly vary between reactants and products. The bias has the form $V_i(q) = 1/2 K_i (q - q_{0,i})^2$, where $K_i$ is the force constant of the $i$th umbrella and $q_{0,i}$ its center. An MD simulation is then performed within each umbrella. To study the proton hopping, the chosen collective variable is a difference of coordination numbers (CNs) between the BAS and the two oxygens involved in the hopping:

$$q = CN(O_i; H) - CN(O_j; H) = \frac{1 - \left(\frac{r_{O_iH}}{r_0}\right)^N}{1 - \left(\frac{r_{O_iH}}{r_0}\right)^{2N}} - \frac{1 - \left(\frac{r_{O_jH}}{r_0}\right)^N}{1 - \left(\frac{r_{O_jH}}{r_0}\right)^{2N}} \quad (2)$$

The specific values of the $N$ and $r_0$ parameters were adapted based on the reaction conditions, more information can be found in Supplementary Note 5.2. The bias potential was applied using PLUMED[55,56] and the final statistical analysis of the data was performed with our in-house developed ThermoLIB library[57]. For some of the hoppings, additional wall potentials were required to prevent undesired side reactions; further details are reported in Supplementary Note 3.1.4

### DFT molecular dynamics

To perform the DFT MD simulations, the CP2K software package (version 7.1)[58,59] was employed to compute energies and forces at a revPBE-D3/TZVP[60–62] level of theory. Because of the mixed plane waves −atom-centered orbitals approach[63] used by CP2K, the plane waves energy cutoff was set to 350 Ry and GTH pseudopotentials[64] were used to smooth the electron density in the proximity of the nuclei. A significant dependency of the forces on the plane waves cutoff was found, but this was shown to have a negligible impact on the final FESs when much higher settings are used (Supplementary Note 3.1.5). The time step for the integration of the equations of motion was set to 0.5 fs. After equilibration of the unit cell (Supplementary Note 3.3.1), production runs were performed in the NVT ensemble using a Nosé-Hoover thermostat with a chain consisting of five beads[65,66] to control the temperature and a time constant of 334 fs (100 cm$^{-1}$).

### MLP training and usage

A SchNet MLP was trained with the SchNetPack package on the DFT energies and forces which were extracted every 5 fs from the DFT US simulations at 873 K[28,29]. First, the energies and forces were unbiased by subtracting the bias potential applied in the US simulations with PLUMED[55,56]. The unbiased DFT datapoints were randomly divided in a training and validation set with a 80:20 ratio. Subsequently, the MLP was trained with a cutoff of 6 Å, 128 features, 50 gaussians and 6 interaction blocks. The resulting MAE on the validation set is 41.9 meV/Å. More details on the training are provided in Supplementary Note 5.1. Classical unbiased and US simulations with the trained MLP were performed with our in-house code YAFF[67] using a time step of 0.5 fs and a Nosé-Hoover thermostat with three beads for temperature control[65,66]. PIMD simulations were performed with the i-PI driver[68] using a time step of 0.25 fs and a PILE thermostat[69] with a time constant of 100 fs for temperature control. Because of the harmonic repulsion between the beads, some of them might explore regions of the phase space that are not necessarily well-sampled in classical DFT US. Therefore, we also performed an extra DFT PIMD US simulation for the 2−3 hopping (Supplementary Note 7.2) and the resulting FES shows an excellent agreement with the MLP one. It is important to remark that this agreement is very likely not generalizable to other systems or reactions and should always be tested appropriately[70]. In both the classical and PIMD US simulations, PLUMED was used to apply the bias.

## Kinetic rate constant calculation

The plain activation free energy obtained from a FES is largely dependent on the choice of collective variable[31,32]. To remove such dependency, it is necessary to move towards a more general macroscopic property of the process under study, namely the kinetic rate constant. In the Bennett-Chandler approach to transition state theory[34,35], the rate constant of a reaction can be written as[71]:

$$k_{BC}(t) = \langle \dot{q}(0)\theta(q(t) - q^*) \rangle_{q(0)=q^*} \frac{e^{-\beta F(q^*)}}{\int_{-\infty}^{q^*} e^{-\beta F(q)}dq},\qquad(3)$$

where the first term is the ensemble average of the time derivatives of $q$ for trajectories that, starting atop the transition state ($q(0)=q^*$), end up in the product basin (as imposed by the Heavyside function $\theta(q(t) - q^*)$). With the MLP, it is possible to explicitly compute the first term by performing a large number of unbiased MD simulations (5000 in this case) starting on the transition state and monitor how many of them effectively end up in the product basin[27,33]. The true rate constant is, in principle, given by $k_{BC} = \lim_{t \to +\infty} k_{BC}(t)$. Luckily, its value quickly reaches a plateau and 50 fs of simulation were sufficient to obtain well-converged results (Supplementary Note 6.2). The rate constant calculated in this manner is referred to as the Bennett-Chandler one ($k_{BC}$).

In general, this approach is too expensive, especially for the DFT case, so that only the approximate transition state theory constant ($k_{TST} = \lim_{t \to 0^+} k_{BC}(t)$) can be computed from the US trajectories, thereby avoiding the need for additional simulations. While $k_{TST}$ represents an upper limit of the true kinetic constant, assuming a recrossing probability equal to zero, it can be used to compare the DFT and MLP results. Further details are reported in Supplementary Note 3.3. To calculate the quantum rate constants, taking NQEs into account, the approximate technique of RPMD was used (see Supplementary Note 7.4)[72]. Although this approximation can only capture short-time quantum effects, it has been shown to yield good quantum rates in comparison with other approximations[73] or quantum mechanical calculations[74].

As the kinetic constant values can span several orders of magnitude, we often make use of Eyring's equation to convert them into phenomenological barriers, which encode the same information while being – in our opinion – more tangible than a reaction rate.

$$\triangle F_{phen}^{\ddagger} = -\frac{1}{\beta}\ln(\beta h k)\qquad(4)$$

## Data availability

The complete training set, examples of input files, processing scripts and the trained MLP have been deposited in the Zenodo database (https://zenodo.org/record/7267913#.Y2U8tHbMK3A). Any additional data is available from the authors upon request. An extended discussion of the results can be found in the Supplementary Information. Source data are provided with this paper.

## Code availability

CP2K (https://github.com/cp2k/cp2k), PLUMED (https://github.com/plumed/plumed2), SchNetPack (https://github.com/atomistic-machine-learning/schnetpack) and YAFF (https://github.com/molmod/yaff) are all open source and freely available at the provided links. ThermoLIB is available upon request at https://molmod.ugent.be/software/thermolib.

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

## Acknowledgements

This work was supported by the Fund for Scientific Research Flanders (FWO, BioFact Excellence of Science project G0H0918N, ID EOS: 30902231, and project G024019N), the Flanders Industry Innovation Moonshot (ARCLATH II, No. HBC.2021.0254) and the Research Board of Ghent University (BOF). The computational resources (Stevin Supercomputer Infrastructure) and services used in this work were provided by the VSC (Flemish Supercomputer Center), funded by Ghent University, FWO, and the Flemish Government – department EWI.

## Author contributions

M.B. and V.V.S. initiated the discussion and designed the scope of the project. M.B. performed the classical DFT simulations. A.L. provided technical support for the PIMD simulations and performed the PIMD DFT simulations. Under the supervision of T.V., R.G. trained the MLP and performed the related simulations, with support from M.C.C. M.B., R.G. and A.L. analyzed the results, which were discussed among all authors. M.B. and V.V.S. wrote the manuscript, with contributions from all authors.

## Competing interests

The authors declare no competing interests.
