## [Peer Review File · Nature Communications]

Reviewer comments, first round –

Reviewer #1 (Remarks to the Author):

Please see attached pdf

Reviewer #2 (Remarks to the Author):

Quantum effects of proton hopping in zeolite chabazite are systematically studied in this work by an acceleration of computational model calculations. To this end, a machine learning approach is employed so that the quantum effects can be studied by path integral molecular dynamics. Classical and quantum particle free energy surfaces for the proton dynamics between oxygen atoms on the zeolite surface are compared. The authors show that activation barriers are affected by quantum effects for relatively high temperatures up to 473 K, whereas they are of lesser importance at higher temperatures. The formation of intra-framework interactions of the Bronsted protons appear to be also affected.

The work is of high technical quality, and these results are highly important and will certainly be of interest for future studies of proton motion. The paper has the potential to generate a high impact in catalytic science using zeolites. It is fun to read. However, before a clear recommendation can be made to accept this paper, the authors should be given the opportunity to answer the following questions or comments:

1.) Catalytic reactions with zeolites are often carried out at temperatures higher than 473 K, so the authors should explain, which type of reactions they think are candidates to be affected by this.

2.) The activation barrier for the proton hopping process is 55.8 kJ/mol (Figure 7a), and it is 11 kJ/mol higher without quantum effects. Previous calculations without considering the quantum nature of the protons generally yielded higher activation barriers than reported here (Figure 1b, CHA). However, experimental values from the literature are lower. So, there is a discrepancy between the value obtained here and the experimental data in the literature. The authors claim that it is almost impossible to obtain a fully dehydrated zeolite, and this seems to be the explanation they offer. A solid-state NMR paper by Huo et al is cited to make this statement (reference 19). Those authors indeed declare that they had water contamination. It is absolutely important for ^1H solid-state NMR of zeolite acid sites to employ an advanced sample handling procedure, and this has been shown frequently. On the other hand, Franke and Simon have shown (by impedance measurements) that the proton conductivity only goes up dramatically, when there is a water chain between two sites to allow the Grotthuss mechanism. Another question is, how trace amounts of water that are far away would affect local proton hopping. In-situ infrared spectroscopy is somewhat easier because those experiments use self-supporting zeolite wafers in an in-situ cell, where the zeolite can be dried while studied. For example, a more recent paper by Losch et al. (JACS 2018) reports an activation barrier of 40 kJ/mol. Therefore, after decades of experimental work on zeolite acid sites, this seems to be a sweeping judgement to say that none of those materials was sufficiently dry to hold up with these new calculations. Experimental work – by nature – deals with real materials, and yes, it is true, sometimes with water contaminations, but there are other factors, such as Si/Al ratio or defect sites, or site distributions. Therefore, there is a lack of an equal and fair comparison of the results for real materials and the ideal model structure used in this work.

Reviewer #3 (Remarks to the Author):

Bocus et.al. performed several molecular dynamics simulations including ab initio molecular dynamics (AIMD), path integration molecular dynamics simulation (PIMD), machine learning

potential (MLP) based molecular dynamics simulation combined with enhanced sampling to explore the nuclear quantum effects (NQEs) on the proton transfer process within Brønsted acid site of H-CHA zeolites over a wide range of temperature. The results show that the quantum nature has a significant effect, which may reduce the proton hopping barrier for more than 10 kJ/mol. Most importantly this work shows that, on the basis of computational speed-up MLP, which was trained against a large number of density functional theory (DFT) structures, it is possible to simulate the acid-catalyzed reactions in zeolites with the timescale dramatically expanded from ps to μ s with the accuracy almost matching with DFT calculations. It can be thought that these computational protocols presented in this work would be very interesting and helpful in the zeolite catalysis community of theoretical modeling. Therefore, I would like to recommend to publish this work after the following technical points are clearly explained in details.

- 1) The simulation results of the CHA zeolite at a very high temperature (873K) was used in the MLP training. However, the high temperature systems usually induce the large deformation of structure, and miss the subtle basins of potential surfaces. As a result, it may affect the accuracy of proton hopping barrier. Some simulations at low temperature range should be added. The proton hopping can also be observed with the umbrella sampling technique.
- 2) The authors argued the very low experimental barriers for the proton hopping are most likely related to the humid environment of zeolites. Does the MLP work for this case? For example, put a couple of water molecules around the BAS to study its effects on the barrier of proton hopping.
- 3) The reason for the high temperature leading to large barrier (as shown in Figure 3 and Figures 15-16) should be clarified properly.
- 4) Figure S8 shows that the force MAE goes down with the increasing training time. However, it looks like that it is acceptable after 30h. Did the authors consider the over training problem in the machine learning process?
- 5) Can the trained MLQs be tabulated as separately files? Hence, these file can be channeled to other MD engines such as LAMMPS. At least, these potential files should be provided in the supporting information.
- 6) In the MLQs based molecular dynamics simulations, is there a scheme to pick outliers which is not sampled in the AIMD/PIMD and appears in the MLQs based molecular dynamics simulations? Is the MLQs refined in the MLQs based molecular dynamics simulations?

Typos and questions:

- 1) It seems that Figure 5 is not necessary in the main text.
- 2) Figure S3, the "H-SSZ-13" is inconsistent with that shown in other places.
- 3) Figure S9, x-label ($d(H,O_i)$) and y-label($d(H,O_j)$) is inconsistent with that in Eq. S5.9. they should be $r(H,O_i)$ and $r(H,O_j)$.

Reviewer #4 (Remarks to the Author):

The work studies the proton hopping kinetics in zeolite catalysis via a combined approach of ML potentials, Umbrella Sampling, and PIMD. In particular, it aims to tackle one of the missing links between simulation and experiment in the description of the activation energy of proton hopping in zeolites, namely the inclusion of NQEs. Since PIMD requires running multiple replicas, it is computationally prohibitive. The authors propose to leverage the improved computational efficiency of ML force-fields for this task.

The work is a good solution to an important problem and I believe deserves publication in this venue. A few more detailed points below:

1. It is unclear to the reader why high-temperature simulations were performed at 873 K, what is special about this temperature?
2. This sentence is a bit disturbing:

" Surprisingly, not only did none of the MLP simulations presented

obvious instabilities, but the error on the forces also remains moderate for frameworks that do not share any secondary building unit with CHA, varying between $196 \text{ meV}\cdot\text{\AA}^{-1}$ for MOR and $258 \text{ meV}\cdot\text{\AA}^{-1}$ for MFI. T"

Not blowing up should hardly be considered a success for a ML potential and errors of $200 \text{ meV}/\text{\AA}$ are quite high and may even lead to qualitatively wrong results. RDFs are a nice unit test but also usually fairly easy to get right. While improving this may require more fundamental advances in ML potentials, I believe the language should be adjusted here.

3. Reproducibility: it would be great to give details on the Nose-hoover thermostat, in particular the Nose mass.

4. Data availability: this work has amassed a large amount of DFT data, these should be publicly shared and documented well, not only be made available upon request! The same goes for the trained potential files and all input files. Much of the progress achieved in this work would be put to waste if these data are not published.

In their submitted paper, Bocus et al explore via computation the possibility of nuclear quantum effects (NQE) influencing the kinetics of proton hopping in zeolites. The topic is an interesting one, and the use of machine learning to train a potential energy function is part of an ongoing “wave” in theoretical chemistry and related fields. The central question here is whether this paper is novel enough or of enough broader interest to justify publication in *Nature Communications* versus in a more specialized journal like *Journal of Chemical Theory and Computation* or *Journal of Chemical Physics*. After studying this paper at length and comparing it to the current state of the art to the best of my abilities, I am sorry to say that I have concluded that this submitted paper is not appropriate for *Nature Comm* and would be better suited for a more specialized journal. My primary reasons are as follows:

- (1) The authors go to considerable length via computational means (see comments below) to elaborate the possibility of NQEs affecting the rates and Arrhenius-like behavior of proton hopping across several sites in zeolites. They conclude that at room temperature and even at higher industrially relevant temperature NQEs may be important, i.e., up to a factor as large as ten or more at the lower temperatures. The issue here, however, is that a NQE (the quantum enhancement relative to classical mechanics) is not an experimentally measurable quantity since classical mechanics does not exist in reality. What is a measurable quantity would be a kinetic isotope effect (substitution of H by D), but these authors do not seem to calculate this. In turn, this effect is something that could be measured experimentally in principle. As such, these calculations are more of a novelty of interest to the theory community but not of relevance to a broader audience because the NQEs cannot be measured directly (it is not possible).
- (2) Another angle that might justify publication in *Nature Comm* would be the novelty of a computational approach, in this case the use of machine learning to train a potential energy surface from electronic DFT calculations, and to then use that ML surface in more extensive simulations (in this case path integral MD, or PIMD). Unfortunately, this idea is not new and it is being done quite a bit (recent work by Roberto Car on liquid water published in PNAS comes to mind). Moreover, by using a ML potential in another way, it is now possible to do highly efficient direct ab initio MD (AIMD) calculations with PIMD for the NQEs [see Li and Voth, *J. Chem. Theory Comp.* **18**, 599-604 (2022)]. This was not done.
- (3) Additionally, there are questions of accuracy in the computational ML potentials fit to DFT. It seems the error between the DFT and the ML potential derived barriers (Fig 4) are about $2 k_B T$ which is not negligible in an exponential Arrhenius factor (a factor of ~ 7 in rate), perhaps giving an error as large as the NQE itself in some cases. Also, the ML potential is trained to classical DFT (AIMD) calculations at a high temperature. It not clear that classical simulations at a high T explore some of the regions of the potential that a nuclear quantized system would do (classically “forbidden” regions), so it is not

clear that the ML PIMD would give the same behavior as *ab initio* PIMD. For liquid water this is not the case that classical high-T is like a NQE at lower-T (see C. Li, et al, , J. Chem. Theory Comp. **18**, 2124-2131 (2022)).

- (4) Two other points of clarification. First, the idea of using the imaginary time Feynman path centroid in a quantum TST as done in the authors' Eq. (S7.17) certainly did not originate with them nor with Manolopoulos. The earliest reference in that regard would be Voth, et al, J. Chem. Phys. **91**, 7749-7760 (1989) which should be cited along with some related earlier work by Gillan cited therein. Secondly, for calculating the quantum "dynamical" effects via a classical-like Bennett-Chandler formula, the authors appear to use the ad hoc RPMD approach. Here it should be clarified that RPMD is ad hoc and has no real theoretical justification in terms of actual quantum dynamics [see S. Jang, A. V. Sinitskiy, and G. A. Voth, J. Chem. Phys. **140**, 154103 (2014)]. PS-The authors should also not claim that the so-called "Matsubara dynamics" of Althorpe somehow "derives" RPMD as it does not. Matsubara dynamics is also quite approximate and one cannot really derive one approximation from another.

In their submitted paper, Bocus et al explore via computation the possibility of nuclear quantum effects (NQE) influencing the kinetics of proton hopping in zeolites. The topic is an interesting one, and the use of machine learning to train a potential energy function is part of an ongoing “wave” in theoretical chemistry and related fields. The central question here is whether this paper is novel enough or of enough broader interest to justify publication in Nature Communications versus in a more specialized journal like Journal of Chemical Theory and Computation or Journal of Chemical Physics. After studying this paper at length and comparing it to the current state of the art to the best of my abilities, I am sorry to say that I have concluded that this submitted paper is not appropriate for Nature Comm and would be better suited for a more specialized journal. My primary reasons are as follows:

We would like to thank the reviewer for the thorough evaluation of the manuscript. The reviewer’s comments were very useful and helped us to substantially improve the paper. While the reviewer appreciated the research performed, he/she concluded that the submitted work was not novel enough to warrant publication in a multidisciplinary journal like Nature Communications. Inspired by the reviewer’s comments, we were stimulated to thoroughly reflect on the added value of this work within the current available literature and to perform additional simulations with the objective of making the message of the paper even stronger. One of the very important suggestions of the reviewer concerns the study of the kinetic isotope effect. As will become clear from the further detailed responses, we performed additional work which showcases how the MLP can also be used to effortlessly compute the reaction KIE, making the key message of the paper even stronger.

Before going in depth to the individual comments of the reviewer, we resume hereafter the most compelling novel points of the presented work.

The derivation of a deep neural network Machine Learning Potential (MLP) for activated events in zeolite catalysis has, to the best of our knowledge, not been reported so far. Nevertheless, one could argue that all the tools to achieve this result are available in literature. Therefore – in our opinion – the most significant contribution of this manuscript are not the individual methodological pieces, but the key scientific advances within the field of theoretical catalysis that have become possible thanks to the availability of a reliable deep neural network MLP. The key improvements, which were previously unattainable using first-principles methods, are summarized in the following three points:

- (i) Much longer molecular dynamics simulations become feasible, giving rise to better converged statistic properties (e.g. free energy profiles) for reactive processes within zeolite catalysis. It is well known that the length of the MD simulations at the DFT level are a weak point in the derivation of thermodynamic and kinetic properties. In total here we performed simulation of ca 0.6 μ s, which is far beyond any standard DFT approach.
- (ii) We were able to account for NQEs in a rigorous way based on PIMD simulations. To the best of our knowledge this contribution is the first application of PIMD/MLP to an activated event in heterogenous catalysis.
- (iii) Accurate reaction kinetics could be determined beyond classical transition state theory, through the reaction flux formalism which also allows barrier recrossing. So far, such approach was completely unfeasible within heterogeneous catalysis because of its computational cost, as thousands of short unbiased MD trajectories must be performed starting atop the transition state region. Albeit various seminal papers appeared in the literature concerning methods for quantum rate computations, to the best of our

knowledge this is the first time these are used for a complex catalytic system such as the one investigated here. More details are provided in the replies to comment 2 and 4.

Hereafter we respond in detail to all comments and questions of the reviewer.

- (1) The authors go to considerable length via computational means (see comments below) to elaborate the possibility of NQEs affecting the rates and Arrhenius-like behavior of proton hopping across several sites in zeolites. They conclude that at room temperature and even at higher industrially relevant temperature NQEs may be important, i.e., up to a factor as large as ten or more at the lower temperatures. The issue here, however, is that a NQE (the quantum enhancement relative to classical mechanics) is not an experimentally measurable quantity since classical mechanics does not exist in reality. What is a measurable quantity would be a kinetic isotope effect (substitution of H by D), but these authors do not seem to calculate this. In turn, this effect is something that could be measured experimentally in principle. As such, these calculations are more of a novelty of interest to the theory community but not of relevance to a broader audience because the NQEs cannot be measured directly (it is not possible).

We agree with the Reviewer; as NQEs are inherently part of nature, the difference between a computational inclusion of NQEs versus a classical treatment cannot be experimentally measured. An important reason for reporting the significant increase in rate constants due to NQEs is to showcase how the traditional approach based on classical nuclei can lead to significant underestimation of the reaction kinetics up to temperatures that are of catalytic interest and, therefore NQEs should also be considered within the catalysis community when reactions conducted at milder temperatures are studied.

On the other hand, the reviewer has proposed the excellent idea of evaluating the kinetic isotope effect for the reaction when going from protium to deuterium, to strengthen the relevance of this paper for the experimental community. With the trained MLP, it becomes possible to directly evaluate the KIE for the hopping reaction from a theoretical standpoint by substituting protium with deuterium (or, in other words, integrate the equations of motion with the mass of the hydrogen atom set to 2.014 a.m.u.). This is possible since the underlying potential energy surface, in principle, does not depend on the mass of the nucleus, but only on its charge.

We performed additional MLP PIMD simulations of all the hoppings with deuterium at 273, 573 and 873 K. To the best of our knowledge, this approach has so far only been applied to small molecules in gas phase (see *e.g.* Liu & Li *Phys. Chem. Chem. Phys.* **2020**, 22, 344-353).

To ensure that the new simulations with deuterium did not visit novel regions of the phase space where the MLP could have a degraded accuracy, 1000 structures were extracted from the MLP simulation of the 2-3 hopping at 873 K and the error on the forces was benchmarked against DFT. The results are shown in **Figure R1**, where it can be seen how the error remains low over the whole CV range and it is similar to those previously reported for protium (see **Figure S14c**), indicating that the deuterium simulations do indeed explore a similar phase space region as the protium ones.

Figure R1. Mean Absolute Error (MAE) on the MLP forces compared to DFT as function of the CV for the 873 K simulation of the 2-3 hopping with deuterium.

For a complete overview of the results, we refer the reviewer to the new **Section S7.5** of the **Supplementary Information**. Since we believe that these findings are of great interest, they were extensively integrated in the main manuscript. All sections were at least partially adapted to accommodate the new results and the deuterium kinetics together with the KIE as a function of temperature are reported in a new version of **Figure 7** in the main manuscript.

To summarize, we found that the substitution of protium with deuterium leads, as expected, to a primary kinetic isotope effect, with consequent reduction of the hopping rate. Increasing the temperature also leads to a decrease in the computed KIE (from ~ 3 to ~ 1.4), in line with the expectations from Bigeleisen-Mayer theory [for an overview see Bigeleisen & Wolfsberg, *Adv. Chem. Phys.* **1958**, 1, 15-76].

Unfortunately, there has not been any attempt so far to measure the kinetic isotope effect for proton hopping in zeolites, to the best of our knowledge. Therefore, the results will have to await experimental validation but the limited effect of deuterium substitution above room temperature seems to be in line with previous static computational investigations performed on cluster models and based on semiclassical transition state theory [Fermann & Auerbach *J. Chem. Phys.* **2000**, 112, 6787-6794].

We thank again the reviewer for the suggestion on the KIE addition. We believe that with this addition we have showcased for the first time how an MLP trained only on classical simulations of the hopping process in a highly relevant heterogeneous catalyst can be used for an accurate PIMD characterization of the reaction. Moreover, the results are now also of broader interest to the experimental community, providing for the first time a computational estimate of KIE for the proton hopping reaction.

(2) Another angle that might justify publication in Nature Comm would be the novelty of a computational approach, in this case the use of machine learning to train a potential energy surface from electronic DFT calculations, and to then use that ML surface in more extensive simulations (in this case path integral MD, or PIMD). Unfortunately, this idea is not new and it is being done quite a bit (recent work by Roberto Car on liquid water published in PNAS comes to mind). Moreover, by using a ML potential in another way, it is now possible to do highly efficient direct ab initio MD (AIMD) calculations with PIMD for the NQEs [see Li and Voth, *J. Chem. Theory Comp.* **18**, 599-604 (2022)]. This was not done.

We agree that some pieces of the presented approach have been published, albeit never in the field of zeolite catalysis or heterogeneous catalysis. More information on the novelty of the approach have been given in the introduction of this rebuttal. More specifically in reply to this comment, recent papers have used enhanced sampling techniques to train reactive MLPs (as an example, Devergne et al. *J. Chem. Theor. Comput.* **2022**, 18, 5410-5421) and MLPs have been used to include NQEs (like in the work of Car, as mentioned by the Reviewer). However, to the best of our knowledge, this is the first time that enhanced sampling in combination with MLPs, are applied to obtain accurate quantum kinetic constants for reactive events in zeolite catalysis beyond the commonly used transition state theory. We showcase a methodology that is readily applicable to any activated event and the reaction we used as case study is highly relevant in the catalysis community. The presented work is a showcase example of a methodology which opens promising perspectives for a field where the inclusion of NQEs is basically non-existent.

Concerning the paper of Li and Voth cited by the reviewer, we were familiar with their work, as it was already cited in the original submission of the paper in the context of techniques that allow to mitigate the huge computational cost of PIMD simulations. As the reviewer states, the simulations in the work of Li and Voth still rely on ab initio forces and energy evaluations and it is about 10^2 times faster than full ab initio PIMD. In our case the MLP itself is used to propagate the system giving an increase of about 10^4 times compared to DFT, while maintaining a similar accuracy as DFT. Given the training protocol we introduce for the MLP, we show that it is possible to obtain an MLP with DFT accuracy for a challenging system such as the chabazite zeolite, which allows to eliminate the ab initio monitoring of the MLP in Voth's method. Even if the 16 beads have to be evaluated one-by-one, we would still retain a speedup of 2-3 orders of magnitude compared to Voth's simulations. Of course, in a very low-temperature regime where perhaps hundreds of beads are needed to obtain converged results, the path integral acceleration techniques such as ring polymer contraction (RPC) or multiple time stepping (MTS) can be used, just as in Voth's method. Since for our application area, namely heterogeneous catalysis, temperatures do not become extremely low to motivate an enormous expansion of beads, such further acceleration schemes were not necessary. On the other hand, we believe that the concept could be inspiring for the catalysis community if the methodology we present would be applied to processes conducted at cryogenic temperatures. Therefore, we included the following sentence to the **Discussion** section:

We remark that for simulations at cryogenic temperatures the number of beads required to achieve converged results could become very large even for the MLP. This problem can be mitigated by coupling the MLP simulations with path integral acceleration techniques.²¹

Where we refer to the review paper of Markland and Ceriotti since it provides a quite comprehensive overview of the available techniques.

(3a) Additionally, there are questions of accuracy in the computational ML potentials fit to DFT. It seems the error between the DFT and the ML potential derived barriers (Fig 4) are about 2 kBT which is not negligible in an exponential Arrhenius factor (a factor of ~ 7 in rate), perhaps giving an error as large as the NQE itself in some cases.

For some of the phenomenological barriers there is indeed some deviation between the MLP and DFT results. We must point out, however, that free energy surfaces obtained from enhanced sampling techniques generally suffer from two separate sources of error. There is i) an error on the PES calculation and ii) an error due to an incomplete sampling of the PES. While the reviewer's comment concerns the first point as a potential source of errors, we believe that the main problem is the second. This is particularly clear by looking at the old **Figure 3** of the main manuscript (also copied below in

Figure R2), where the relatively high barrier at 573 K obtained for the DFT case is caused by the spike around the transition state. Such free energy surface (FES) fluctuations are not so uncommon in umbrella sampling and are caused by the system jumping from one side to the other of the transition state in a relatively slow fashion, causing issues with the simulation's ergodicity. At higher temperatures (like in the 873 K case) or with more umbrellas (like in the 273 K case, see Table S2) the problem is indeed absent.

Because of the massive computational speedup offered by the MLP, we could easily use more umbrellas (twice as many were used) with tighter force constants and simulate for a much longer time (100 ps for the MLP versus 50 ps for DFT), providing extremely well-converged FESs. To demonstrate this, we considered the problematic 573 K DFT simulation of the 2-3 hopping and added an additional 40 ps of simulation time to each umbrella (for a total of 90 ps per umbrella). Moreover, 2 additional umbrellas were added at CV=-0.05 and CV=0.05 with a force constant of 1500 kJ·mol⁻¹. With this large increase in the sampling, the spike around the TS – as expected – disappears and excellent agreement between the MLP and DFT results is obtained. This can nicely be visualized in the new Figure 3 and Figure 4 of the main manuscript. For the reviewer's convenience, we report the changed figures side-by-side in Figure R2.

Figure R2. Comparison between the previous and current versions of Figure 3 and Figure 4 in the main manuscript. Remark that in Figure 4 there are two additional points related to additional 273 K simulations for the 1-4 hopping that were performed following the suggestion of Reviewer 3 (comment 1).

In our opinion, these results provide further strength to the methodology, highlighting how the redundancy of the phase space explored during the computationally cheap MLP MD simulations can be effectively exploited to shorten the required expensive DFT sampling and effectively converge the FEPs with the MLP.

To highlight these results, we included the following additions to the main manuscript in the **Introduction** (the text in light blue was already present):

...which allowed to: (i) compute the Free Energy Surfaces (FESs) of all possible hoppings around an isolated Al defect in the temperature range 273-873 K with a large number of umbrellas and long simulation times to obtain well-converged results, (ii) explicitly...

And in the **Construction of a reactive MLP with DFT accuracy for proton hopping** section:

Initially, the 573 K DFT profile presented a moderate spike in the transition state region, which was not present in the MLP profile. Therefore, 2 additional umbrellas were added in the proximity of the transition state and an additional 40 ps of simulation was performed in every umbrella, for a total of 90 ps. The final DFT profile reaches almost perfect agreement with the MLP one, highlighting how the (small) differences between MLP and DFT FESs are almost certainly caused by sampling issues rather than by significant deviations in the underlying PES.

(3b) Also, the ML potential is trained to classical DFT (AIMD) calculations at a high temperature. It not clear that classical simulations at a high T explore some of the regions of the potential that a nuclear quantized system would do (classically “forbidden” regions), so it is not clear that the ML PIMD would give the same behavior as ab initio PIMD. For liquid water this is not the case that classical high-T is like a NQE at lower-T (see C. Li, et al, J. Chem. Theory Comp. 18, 2124-2131 (2022)).

We agree with the reviewer that there is no a priori guarantee that the phase space visited during PIMD simulations is adequately explored during high T classical simulations, and this was indeed a concern we had from the beginning. To investigate this possible discrepancy, we already presented in the original SI a complete free energy surface for the 2-3 hopping at 273 K using DFT PIMD (see **Section S7.2** of the **Supporting Information**). Unfortunately, the prohibitively high simulation cost did not allow us to obtain a well-converged FES, but the agreement with the MLP results remains quite remarkable (**Figure S14a**). Moreover, to demonstrate beyond doubt that no such forbidden regions occur for our system which the MLP would fail to accurately capture, snapshots from the DFT PIMD simulation were extracted. From these, the errors on the forces predicted by the MLP were computed (see **Figure S14c**). All errors are similar to those seen during the validation of the classical MD simulations, demonstrating that no regions of the phase space become accessible which the MLP cannot accurately capture.

Therefore, we believe we demonstrated that the MLP simulations do not suffer significantly from this type of problem. We do not claim that this is a feature of all possible reactions in zeolites and there will certainly be problematic cases for which this will not be the case. Would that be the case, control procedures to include not-yet-sampled structures (like the query by committee method) can be easily applied. Moreover, given the very short trajectories required to train more modern networks (see **Section S10** in the **Supplementary Information**) short DFT PIMD simulations could be directly used to generate the training set.

To hopefully better clarify this point while trying to keep the length of the paper acceptable, we propose the following adaption to the *MLP training and usage* paragraph of the **Methods** section, where we also refer to the interesting paper from Voth’s group mentioned by the reviewer:

...and the resulting FES shows an excellent agreement with the MLP one. It is important to remark that this agreement is very likely not generalizable to other systems or reactions and should always be tested appropriately.⁷¹

- (4) Two other points of clarification. First, the idea of using the imaginary time Feynman path centroid in a quantum TST as done in the authors' Eq. (S7.17) certainly did not originate with them nor with Manolopoulos. The earliest reference in that regard would be Voth, et al, *J. Chem. Phys.* 91, 7749-7760 (1989) which should be cited along with some related earlier work by Gillan cited therein. Secondly, for calculating the quantum "dynamical" effects via a classical-like Bennett-Chandler formula, the authors appear to use the ad hoc RPMD approach. Here it should be clarified that RPMD is ad hoc and has no real theoretical justification in terms of actual quantum dynamics [see S. Jang, A. V. Sinitskiy, and G. A. Voth, *J. Chem. Phys.* 140, 154103 (2014)].

Following the suggestion of the reviewer, we have added the reference to the 1989 article of Voth regarding the formulation of quantum transition state theory in the corresponding section S7.4 of the SI. We certainly did not intend to imply that the idea of a quantum TST originated from us or Manolopoulos, but merely cited the relevant papers which explain the methodology used to compute the quantum rate constants in this work. The combination of RPMD and the Bennett-Chandler approach to include dynamical recrossing effects in the reaction rate has also been used in other works [Colleparado-Guevara et al. *J. Chem. Phys.* **2008**, 128, 144502; Suleimanov et al. *J. Phys. Chem. A* **2016**, 120, 8488-8502] and gave satisfactory accurate results. As mentioned in the SI, RPMD is indeed an approximate technique to include short-time quantum dynamics. While a rigorous theoretical derivation of the RPMD approximation might be arguable and the limits of the approximation should be kept in mind, its ability to correctly predict rate constants has been shown in several works [Pérez de Tudela et al. *J. Phys. Chem. Lett.* **2012**, 3, 493-497; Menéndez et al. *J. Phys. Chem. A* **2019**, 123, 7920-7931]. Moreover, the article of Voth [Jang et al. *J. Chem. Phys.* **2014**, 140, 154103] indicated by the reviewer also explicitly mentions the agreement between RPMD and exact results for linear operators in position and momentum for the harmonic oscillator, while artefacts of the RPMD approximation are particularly present for nonlinear operators. In this respect, a recent article of Voth [Loose et al. *J. Chem. Theor. Comput.* **2022**, 18, 5856-5863] also showed an excellent agreement between the self-diffusion constants of para-hydrogen and liquid water calculated using RMPD and centroid molecular dynamics (CMD), a different approximation to include quantum dynamics. Finally, it should also be pointed out that in the absence of recrossing (the limit for t approaching 0 in Figure S12), the rate as predicted by quantum TST is obtained. And, of course, quantum TST has more elaborate theoretical derivations, as shown for instance by Althorpe and Hele [Hele & Althorpe *J. Chem. Phys.* **2013**, 138, 084108; Althorpe & Hele *J. Chem. Phys.* **2013**, 139, 084115]. This implies that even without considering dynamical recrossing effects, the conclusions of this work regarding the rates would also remain valid within the TST approximation.

To better clarify the approximate nature of RPMD, we adapted the **Methods** section with the following addition:

*To calculate the quantum rate constants, taking NQEs into account, the approximate technique of RPMD was used (see **Section S7.4** of the **Supplementary Information**).⁷³ Although this approximation can only capture short-time quantum effects, it has been shown to yield good quantum rates in comparison with other approximations⁷⁴ or quantum mechanical calculations.⁷⁵*

PS-The authors should also not claim that the so-called "Matsubara dynamics" of Althorpe somehow "derives" RPMD as it does not. Matsubara dynamics is also quite approximate and one cannot really derive one approximation from another.

As this work relies on the RPMD method as known in literature (as explained above) and does not deal with the theoretical derivation of the RPMD approximation, Matsubara dynamics is not mentioned in the manuscript or the SI. Therefore, we refrained from making any claim regarding the theoretical

origin of RPMD and restrict ourselves to mentioning the appropriate references in which the RPMD approximation was introduced.

We thank the reviewer for the in-depth evaluation of the manuscript and constructive remarks made, which helped us to substantially improve the quality of the work. We were also pleased to read the overall evaluation of the manuscript namely that the work is highly important and has the potential to generate high impact in the field of catalytic science using zeolites.

Quantum effects of proton hopping in zeolite chabazite are systematically studied in this work by an acceleration of computational model calculations. To this end, a machine learning approach is employed so that the quantum effects can be studied by path integral molecular dynamics. Classical and quantum particle free energy surfaces for the proton dynamics between oxygen atoms on the zeolite surface are compared. The authors show that activation barriers are affected by quantum effects for relatively high temperatures up to 473 K, whereas they are of lesser importance at higher temperatures. The formation of intra-framework interactions of the Bronsted protons appear to be also affected. The work is of high technical quality, and these results are highly important and will certainly be of interest for future studies of proton motion. The paper has the potential to generate a high impact in catalytic science using zeolites. It is fun to read. However, before a clear recommendation can be made to accept this paper, the authors should be given the opportunity to answer the following questions or comments:

- (1) Catalytic reactions with zeolites are often carried out at temperatures higher than 473 K, so the authors should explain, which type of reactions they think are candidates to be affected by this.

In the previous manuscript version, only a short reference to this aspect was made in the **Full characterization of the hopping kinetics** section:

It appears therefore that for zeolite-catalyzed processes conducted at milder conditions, among which the ones related to biomass conversion are a predominant example,² NQEs might be non-negligible when computing the kinetics of proton-transfer steps.

We agree that this statement was a bit vague and that it is important, certainly for the audience interested on the applicability of the methodology to relevant examples in zeolite catalysis, to be more specific. Since making an exhaustive list of reactions conducted below 473 K would not really be suited for a communication paper, we selected a couple of interesting examples from recent literature of reactions conducted at these conditions that involve hydrogen transfer reactions. The aforementioned paragraph has been amended with the following addition:

A few examples where these effects might be important are the aqueous cyclohexanol dehydration in H-Beta zeolite at temperatures of about 400 K⁴⁰ and the oxidation of methane to methanol in copper-loaded zeolites carried out at ~473 K.⁴¹ These two cases serve as examples of the relevant application area and conditions influenced by NQEs in zeolites. However, it is clear that many more cases could be affected by the inclusion of NQEs.

- (2) The activation barrier for the proton hopping process is 55.8 kJ/mol (Figure 7a), and it is 11 kJ/mol higher without quantum effects. Previous calculations without considering the quantum nature of the protons generally yielded higher activation barriers than reported here (Figure 1b, CHA). However, experimental values from the literature are lower. So, there is a discrepancy between the value obtained here and the experimental data in the literature. The authors claim that it is almost impossible to obtain a fully dehydrated zeolite, and this seems to be the explanation they offer. A solid-state NMR paper by Huo et al is cited to make this statement (reference 19). Those

authors indeed declare that they had water contamination. It is absolutely important for ¹H solid-state NMR of zeolite acid sites to employ an advanced sample handling procedure, and this has been shown frequently. On the other hand, Franke and Simon have shown (by impedance measurements) that the proton conductivity only goes up dramatically, when there is a water chain between two sites to allow the Grotthuss mechanism. Another question is, how trace amounts of water that are far away would affect local proton hopping. In-situ infrared spectroscopy is somewhat easier because those experiments use self-supporting zeolite wafers in an in-situ cell, where the zeolite can be dried while studied. For example, a more recent paper by Losch et al. (JACS 2018) reports an activation barrier of 40 kJ/mol. Therefore, after decades of experimental work on zeolite acid sites, this seems to be a sweeping judgement to say that none of those materials was sufficiently dry to hold up with these new calculations. Experimental work – by nature – deals with real materials, and yes, it is true, sometimes with water contaminations, but there are other factors, such as Si/Al ratio or defect sites, or site distributions. Therefore, there is a lack of an equal and fair comparison of the results for real materials and the ideal model structure used in this work.

We do absolutely agree with the reviewer's statement, as we did not express ourselves well enough in the **Discussion** section. It was not our intention to claim that the experimental barriers are always unreliable because the samples are not handled properly, but to rather state that our model is insufficient to capture the outstanding complexity of the true material. Active site proximity and the presence of defective sites (EFALs, framework-associated aluminum, etc.) were indeed already listed in the **Introduction** as possible causes of discrepancy between our 'perfect' unit cell and the actual material:

However, a more recent investigation has disproven such interpretation and indicated active site proximity effects as the main cause for the observed change in activation energy.¹⁸ Inter-site hopping was also suggested to be responsible for the high activation energies retrieved with IS.⁷

To understand this lack of consistency, it is important to consider the main possible sources of discrepancy between the proton hopping barriers from literature. First, the residual presence of water in the catalyst is often indicated as the main source for the – in general – low experimental barriers,¹⁰ as it is almost impossible to achieve a completely dry material with routine drying procedures.¹⁹ Moreover, the presence of defective sites like extra-framework aluminum species is known to alter the BAS' acidity compared to the pristine material.²⁰

Since the BAS is extremely hydrophilic and it is certain that water massively enhances proton hopping kinetics [see for instance the computational investigation of Liu & Mei, *J. Phys. Chem. C* **2020**, 124, 22568-22576] we assumed that this could be a plausible source of discrepancy.

We were familiar with the very interesting work of Losch to study proton mobility in zeolites through IR spectroscopy. We did not report it as a mean of comparison, however, since the measurements are performed after exposing the zeolite to a 1:1 mixture of H₂O and D₂O. The barriers reported in the study are related to H/D exchange in hydrated conditions, where the authors refer indeed to Grotthuß-like mechanism and hydronium ion diffusion. These types of mechanisms are more related to long-range diffusion rather than local hopping in anhydrous conditions as in our case.

To hopefully better point out the limitation of our model with respect to a realistic material, we amended the **Discussion** section in the following manner:

Lower experimental barriers than the one reported in this work are most likely related to the fact that zeolites are fundamentally never completely dry at experimental conditions and other active sites, like

~~EFAL species, will most likely be present aside pristine BASs~~ The computed activation energy remains relatively higher than the experimentally available ones. The most likely source of discrepancy lies in the perfect crystalline nature of the adopted zeolite model. The presence of residual water molecules, of defects (EFAL species, for instance) and an heterogeneous aluminum distribution are basically unavoidable at the macroscale and all these factors are known to potentially affect the behavior of protons in zeolites.^{18,20,47} ~~Both these factors are known to change the proton acidity and potentially speed up proton transfer reactions.~~^{26,47} According to the simulations, a primary KIE of about 3 is expected at 273 K but no experimental evidence is available thus far to corroborate this result. With the presented proof-of-concept study, it becomes possible to train MLPs for additional reactions and environments, eventually simulating in a more realistic way proton hoppings and activated processes in zeolite catalysis.⁵³

As pointed out in the last sentence of the **Discussion**, we hope that MLPs will allow to simulate more intricate models in the stride of approaching more and more realistic representations of the material used in practical applications, where we also included a citation to the very recent perspective of Ma and Liu (new ref. 53) about the future of MLP for zeolite simulations.

We thank the reviewer for the in-depth evaluation of the manuscript and the interesting comments, which we all addressed and which helped to substantially improve the manuscript. We were very pleased to read the overall assessment of the reviewer namely that the work is very interesting thanks to the dramatic increase in accessible time scales from the ps to the μ s scale and that as such the work would be an interesting contribution for the zeolite catalysis community.

Bocus et. Al. performed several molecular dynamics simulations including *ab initio* molecular dynamics (AIMD), path integration molecular dynamics simulation (PIMD), machine learning potential (MLP) based molecular dynamics simulation combined with enhanced sampling to explore the nuclear quantum effects (NQEs) on the proton transfer process within Brønsted acid site of H-CHA zeolites over a wide range of temperature. The results show that the quantum nature has a significant effect, which may reduce the proton hopping barrier for more than 10 kJ/mol. Most importantly this work shows that, on the basis of computational speed-up MLP, which was trained against a large number of density functional theory (DFT) structures, it is possible to simulate the acid-catalyzed reactions in zeolites with the timescale dramatically expended from ps to μ s with the accuracy almost matching with DFT calculations. It can be thought that these computational protocols presented in this work would be very interesting and helpful in the zeolite catalysis community of theoretical modeling. Therefore, I would like to recommend to publish this work after the following technical points are clearly explained in details.

- (1) The simulation results of the CHA zeolite at a very high temperature (873K) was used in the MLP training. However, the high temperature systems usually induce the large deformation of structure, and miss the subtle basins of potential surfaces. As a result, it may affect the accuracy of proton hopping barrier. Some simulations at low temperature range should be added. The proton hopping can also be observed with the umbrella sampling technique.

This is a concern we also had and was one the main reasons that led us to perform additional simulations at 273 and 573 K for the 2-3 hopping (see **Figure 3**). The results reported in the manuscript show excellent agreement between the low temperature MLP and DFT and, in principle, give confidence that an MLP trained at higher temperatures is also able to reproduce the free energy surfaces at low temperature. However, to be as certain as possible, we performed an additional fully *ab initio* umbrella sampling simulation of the 1-4 hopping at 273 K. This hopping was selected because, together with 2-3 hopping, it is one of the main contributors to the overall hopping rate (see **Figure 7**) and thus we believe it is important to explicitly check the validity of the MLP at lower temperatures for this hopping. Moreover, it allows us to 'probe' all four oxygens at the lowest considered temperature.

The MLP results are again in excellent agreement with the DFT ones, highlighting how it is indeed possible to learn with high accuracy the reaction PES from high-temperature simulations. The comparison between the forward and backward phenomenological barriers for this new hopping has been added to **Figure 4** (black dots around 60-70 kJ·mol⁻¹) and the visual comparison between the free energy surfaces to **Figure S10**. Apart from this extra validations, training on low temperature simulations would likely not bring any substantial improvement to the results because of the limited entropic effects in this reaction, which is something that has also been tested for other cases within our group (see e.g. **Figure 3.3**, pg. 46 of M. Cools-Ceuppens PhD thesis, <http://hdl.handle.net/1854/LU-8766908>). Of course, if the existence of a free energy minimum would largely depend on entropy (possibly completely disappearing at high temperatures), it would then

become important to change the training data generation procedure and sample at multiple temperatures, for instance by using replica-exchange molecular dynamics [Sugita & Okamoto, *Chem. Phys. Lett.* **1999**, 314, 141-151].

Given the cost of performing sufficiently long DFT US simulations, we hope the new results for the 1-4 hopping are sufficient to convince the reviewer of our results' validity, as performing additional benchmark calculations would become extremely expensive. The main text has been amended in various places to refer to the new set of results, the main change being:

*To this end, ~~two~~ three additional sets of DFT US simulations were performed. The 2-3 hopping was tested at 573 and 273 K while the 1-4 hopping at 273 K. Both hoppings have the lowest activation energies and with this choice all four oxygens are considered at the lowest temperature. ~~for the 2-3 hopping were performed at 273 and 573 K. The 2-3 hopping was chosen for most of the benchmarks reported herein, as it is one of the lower activated paths.~~ For the sake of clarity, a detailed comparison between the MLP and DFT results is presented in **Section S6.1** of the **Supplementary Information** while here only the 2-3 hopping is discussed in detail.*

(2) The authors argued the very low experimental barriers for the proton hopping are most likely related to the humid environment of zeolites. Does the MLP work for this case? For example, put a couple of water molecules around the BAS to study its effects on the barrier of proton hopping.

This is a very interesting suggestion; however, we do not believe the MLP is transferable enough to allow for water addition. The currently employed MLP has 'learned' to accurately predict the forces acting on atoms based on the atomic environment around each atom. Therefore, it is familiar with the zeolite OH bond, but this is very different from an OH bond in water (in other words, the MLP has never seen an oxygen bonded only to two protons). This can also be seen when considering other all-silica zeolite topologies, in which the local environment of the Si and O atoms is not so drastically different than the training set, the MLP error deteriorates quite fast with a decreasing degree of similarity between the structures (**Figure S19**). We could therefore perform simulations with water in the zeolite, but without extensive validation the results would be almost certainly completely unreliable.

In principle, it should be possible to train an MLP that can also describe water in zeolites, however this is a study on its own, as we would have to carefully address how to select the additional training data, for which loadings, account for the configurational mobility of water etc. This is certainly a very interesting point; however, we believe that such detailed investigation would lead us quite far from the original scope of the manuscript. The fact that water remarkably reduces the barriers of proton hopping has also already been subject of various studies employing both static [Ryder et al. *J. Phys. Chem. B* **2000**, 104, 6998-7011] and dynamic [Liu & Mei *J. Phys. Chem. C* **2020**, 124, 22568-22576] simulations.

While this perspective on the MLP potentials is likely too detailed for a communication paper, we tried to reformulate the last sentence of the **Discussion** section to imply that the extension of the methodology to more complex environments should be relatively straightforward.

With the presented proof-of-concept study presented, it becomes possible to train more MLPs for other acidic more realistic environments and eventually simulate in a more realistic way proton hoppings and activated processes in zeolite catalysis. This proof-of-concept study presents a general scheme to obtain MLP models that can simulate proton hoppings and activated processes in zeolite catalysis with improved realism. The proposed methodology is, in principle, extendible to additional reactions and reactive environments, making it a valuable tool for studying a wide range of catalytic phenomena.

- (3) The reason for the high temperature leading to large barrier (as shown in Figure 3 and Figures 15-16) should be clarified properly.

The higher reaction barriers are caused by entropic effects. Indeed, the transition state is very rigid, which leads to a negative activation entropy and, consequently, an activation free energy that increases with temperature. This can be visually seen in **Figure S6-S7** of the **Supporting Information** (for the 2-3 hopping) and is present in the static results as well (**Section S2**). From another perspective, in the 2-dimensional plot showing the free energy as function of the distance between the proton and the two framework oxygens involved in the hopping (**Figure S6-S7**) it is clear that a wider range of distances becomes accessible when increasing the reaction temperature, as the proton can move more with an increased thermal energy. When switching to coordination numbers, all these states are 'squeezed' into the tighter reactant/products regions, which results in them to be stabilized with respect to the transition state. A more detailed explanation about the correspondence between distances and coordination numbers can be found in a recent publication from our group [Bailleul et al. *J. Catal.* **2020**, 388, 38-51]. This explanation has now been summarized in the main manuscript as follow (section **Construction of a reactive MLP with DFT accuracy for proton hopping**, p. 6):

...with most variations contained well within the error bars. The free energy barrier exhibits a clear increase with temperature, which is in line with a rigid transition state associated with a negative entropy variation.

- (4) Figure S8 shows that the force MAE goes down with the increasing training time. However, it looks like that it is acceptable after 30h. Did the authors consider the over training problem in the machine learning process?

Overfitting is indeed a concern when training a machine learning model, which can generally be detected by evaluating the MAE of the forces predicted by the MLP on an independent test set.

First, the DFT dataset was randomly split into a training set containing 80% of the data and a validation set containing 20% of the data. The error on the forces reported in **Figure S8** is the error on the validation set (on which no training was performed), not the error on the training set (on which training was performed). If overfitting was present, it would be expected that the validation error would increase during training, which is not observed. However, as correlations can exist between the training and validation set, further independent simulations should be performed to unequivocally demonstrate that the MLP was not overfitted. In **Figure S14c** in the **Supporting Information**, snapshots extracted from DFT PIMD simulation were recomputed with the MLP, demonstrating a MAE on the forces of 42 meV/Å, which is the same as the validation error during training. This independent validation demonstrates that no significant overfitting has occurred during the training of our model.

We agree this was not explained adequately before and, therefore, we added some additional explanation in **Section S5.1** in the **Supporting Information**:

The observation that the validation error does not noticeably increase further on in training yields a first indication that no overfitting of the model occurs. However, as some correlation still exists between snapshots in the training and validation set, this can only be thoroughly confirmed when performing independent simulations. As will be shown later on (see Section S7.2), the MAE on forces of independently performed simulations is 42 meV/Å, confirming the lack of overfitting.

In Section S7.2 in the **Supporting Information**:

The obtained MAE on the forces averaged over all atoms of 42 meV/Å is the same as the validation error of the MLP (see Section S5.1), further demonstrating that no significant overfitting occurred during training.

And in the **Methods** section:

The unbiased DFT datapoints were randomly divided in a training and validation set with a 80:20 ratio. Subsequently, the MLP was trained...

(5) Can the trained MLQs be tabulated as separately files? Hence, these file can be channeled to other MD engines such as LAMMPS. At least, these potential files should be provided in the supporting information.

We would like to apologize with the Reviewers for not sharing the files immediately. We had prepared a folder with all necessary data to reproduce the calculations including the trained potential, but it was mistakenly not uploaded during the first submission. Since, as also suggested by Reviewer 4 (see comment 4), it would be important for reproducibility sake to make the whole training dataset available, we made a freely accessible repository on Zenodo (<https://zenodo.org/record/7267913#.Y2U8tHbMK3A>) and adjusted the **Data Availability** section accordingly.

The complete training set, examples of input files, processing scripts and the trained MLP can be found at <https://zenodo.org/record/7267913#.Y2U8tHbMK3A>. Any ~~other~~ additional data is available from the authors upon reasonable request.

Unfortunately, the potential cannot be straightforwardly converted to tabulated pair potential files for use in LAMMPS, as it is not pair-wise additive but a many-body potential. However, any MD engine which has a python interface (for the PyTorch model) and is coupled with PLUMED (for enhanced sampling) can be used in combination with the model.

(6) In the MLQs based molecular dynamics simulations, is there a scheme to pick outliers which is not sampled in the AIMD/PIMD and appears in the MLQs based molecular dynamics simulations? Is the MLQs refined in the MLQs based molecular dynamics simulations?

Active learning procedures, where the model is iteratively refined to include previously unknown regions of the phase space, are of utmost importance. For instance, some very interesting recent works have shown how techniques such as the query by committee method [Devergne et al. *J. Chem. Theor. Comput.* **2022**, 18, 5410-5421; Erlebach et al. *Npj Comput. Mater.* **2022**, 8, 274] can be effectively used to include structures not yet correctly described by the MLP. We did not adopt such approach here because of the relative simplicity of the reaction under study (remark that in the committee method multiple MLPs have to be trained together and used to evaluate energy and forces, making the whole procedure computationally more expensive). The motions of the proton around the zeolite oxygens are relatively limited and therefore easy to sample. Moreover, using umbrella sampling, each umbrella runs independently from each other starting from different initial velocities, hence making it very unlikely that specific motions in the zeolite framework wouldn't be sampled. Finally, all tetrahedral atoms in the CHA framework are equivalent by symmetry, adding to the redundancy of the data. Given this and the fact that all performed validations did not demonstrate failures of the model to accurately predict the forces for any snapshots, we did not perform any active learning procedure, although the usage of an active learning scheme could be interesting in future applications where one would also be interested in reducing the amount of training data.

Typos and questions:

(1) It seems that Figure 5 is not necessary in the main text.

We agree that the figure does not necessarily provide any particular information, but we feel that it might be interesting – for the broader audience with little familiarity with PIMD – to see how the ring polymer looks like in practice. We would therefore be prone to leave the figure in the manuscript, certainly as this manuscript is under consideration with Nature Communications which aims to address a broader multidisciplinary audience.

(2) Figure S3, the “H-SSZ-13” is inconsistent with that shown in other places.

(3) Figure S9, x-label ($d(H,O_i)$) and y-label($d(H,O_j)$) is inconsistent with that in Eq. S5.9. they should be $r(H,O_i)$ and $r(H,O_j)$.

We thank the reviewer for spotting these two typos, they have now both been addressed.

We thank the reviewer for the in-depth evaluation of the manuscript and the interesting comments and questions. We have taken all comments of the reviewer into account, which allowed to substantially improve the quality of the quality of the work.

The work studies the proton hopping kinetics in zeolite catalysis via a combined approach of ML potentials, Umbrella Sampling, and PIMD. In particular, it aims to tackle one of the missing links between simulation and experiment in the description of the activation energy of proton hopping in zeolites, namely the inclusion of NQEs. Since PIMD requires running multiple replicas, it is computationally prohibitive. The authors propose to leverage the improved computational efficiency of ML force-fields for this task. The work is a good solution to an important problem and I believe deserves publication in this venue. A few more detailed points below:

- (1) It is unclear to the reader why high-temperature simulations were performed at 873 K, what is special about this temperature?

The exact chosen temperature does not have any particular significance. We decided to use 600 °C (873 K) as it is on the higher end of typical process temperatures for zeolite-catalyzed reactions, but using any other temperature as high-limit would likely not change much (as long as the framework does not start to melt, of course).

We clarified this point by adding the following sentence in the main manuscript (section **Construction of a reactive MLP with DFT accuracy for proton hopping**, p. 5):

...where 1 silicon is replaced by Al to give a final Si/Al ratio of 35. The temperature choice of 873 K is arbitrary but, in general, on the higher end of typical zeolite-catalyzed processes.¹

- (2) This sentence is a bit disturbing: "Surprisingly, not only did none of the MLP simulations presented obvious instabilities, but the error on the forces also remains moderate for frameworks that do not share any secondary building unit with CHA, varying between 196 meV·Å⁻¹ for MOR and 258 meV·Å⁻¹ for MFI." Not blowing up should hardly be considered a success for a ML potential and errors of 200 meV/Å are quite high and may even lead to qualitatively wrong results. RDFs are a nice unit test but also usually fairly easy to get right. While improving this may require more fundamental advances in ML potentials, I believe the language should be adjusted here.

We agree with the reviewer's comment on the formulation used and corrected it appropriately as explained below. Initially, we were very pleased with the fact that the simulation was not showing clear instabilities even when applied to chemical environments to which the MLP was not trained, and therefore we might have stressed it too much. Moreover, we also read with great interest the very recent work from Grajciar and co-workers [Erlebach et al. *Npj Comput. Mater.* **2022**, 8, 174] where a general and transferable MLP is constructed for purely siliceous frameworks using an active learning procedure. We tried therefore to be more moderate on our exposition of the results by rephrasing the sentence while also citing Erlebach's paper (the pale text indicates section of text that have not been modified):

Surprisingly, not only did none of the MLP simulations presented obvious instabilities, but the error on the forces also remains moderate for frameworks that do not share any secondary building unit with CHA, varying between 196 meV·Å⁻¹ for MOR and 258 meV·Å⁻¹ for MFI. None of the MLP simulations presented obvious instabilities and the error on the forces is not excessive even for frameworks that do not share any secondary building unit with CHA, varying between 196 meV·Å⁻¹ for MOR and 258 meV·Å⁻¹

¹ for MFI. The quality of the zeolite trajectories, monitored through the Si-O and Si-Si RDFs, remains ~~very high~~ reasonably good with only small long-range differences for MFI (Figure S19). Testing the proton hopping reactivity in a systematic way for more frameworks would require a further set of expensive ab initio US simulations and, therefore, is outside the scope of this work. The results obtained on the all-silica frameworks, nonetheless, still indicate that the MLP can capture in a large extent the chemistry of Si-O-Si bonds and, therefore, we expect that not many additional DFT simulations would be needed to retrain it and extend its accuracy to new zeolite frameworks, for instance building on the transferable MLP for siliceous frameworks by Erlebach et al.⁴² towards aluminum-containing zeolites of catalytic interest.

(3) Reproducibility: it would be great to give details on the Nose-hoover thermostat, in particular the Nose mass.

The requested information on the Nose-Hoover Thermostat was unintentionally left out in the DFT computational details. We report the time constant of the Nosé-Hoover chain rather than the mass (they fundamentally are analogous, see e.g. the original publication from Martyna et al. *J. Chem. Phys.* **1992**, 97, 2635-2643) as this is the input parameter required by CP2K (as before, the pale text was already present):

...using a Nosé-Hoover thermostat with a chain consisting of five beads^{61,62} to control the temperature and a time constant of 334 fs (100 cm⁻¹).

Note that the time constant is not exactly the same as the 100 fs used by the MLP. Nonetheless, based on previous benchmarks within our group, both allow for a good temperature control without coupling with the system's modes. An example of CP2K input file and script to run the MLP simulations is now provided as **Supplementary Material** for the publication, see also next comment.

(4) Data availability: this work has amassed a large amount of DFT data, these should be publicly shared and documented well, not only be made available upon request! The same goes for the trained potential files and all input files. Much of the progress achieved in this work would be put to waste if these data are not published.

We would like to apologize to the Reviewers for not sharing the files immediately. We had prepared a folder with all necessary data to reproduce the calculations, but it was mistakenly not uploaded during the first submission. As rightly pointed out, it would also be important to share the whole training dataset. To this end, we opted to make a freely accessible repository on Zenodo (<https://zenodo.org/record/7267913#.Y2U8tHbMK3A>) where – aside from all the DFT trajectories used to train the MLP – there are examples of input and analysis scripts and the trained MLP. The **Data Availability** section in the main manuscript has also been adjusted accordingly.

The complete training set, examples of input files, processing scripts and the trained MLP can be found at <https://zenodo.org/record/7267913#.Y2U8tHbMK3A>. Any ~~other~~ additional data is available from the authors upon reasonable request.

Reviewer comments, second round –

Reviewer #1 (Remarks to the Author):

I am happy with the revised manuscript and now recommend publication. The calculation of isotope effects adds important data to the paper that can in principle be measured.

Reviewer #2 (Remarks to the Author):

The revised version of this work has addressed my questions, and I now recommend acceptance.

The wording regarding the comparison between experimental work on real materials and computational modeling on idealized structures has been improved, with a prospect for further theoretical work in the area.

I was still hesitating to recommend the paper, because the authors claim, on the one hand, that this is important for catalysis, but then they do not show models that make use of their methods in catalysis. This is a model study of a proton hopping process between oxygen atoms in an AlO_4 site, and I cannot imagine that this would be of relevance in a catalytic reaction. The two examples the authors mention now are the dehydration of an alcohol and methane oxidation on a Cu site. The alcohol OH group and the water product will certainly impact the motion of the zeolite proton, and methane oxidation is a quite different case, because a C-H bond is broken (and not O-H).

However, the finding that quantum effects do play a role at relevant catalytic temperatures in zeolites, and the methods used here are paving the way for future studies in catalysis are deemed highly important by this reviewer. It is the first study of quantum effects of proton hopping in zeolites, to the best of my knowledge, and the revised version now also has added a study on kinetic isotope effects, making the concept amenable for an experimental comparison.

Reviewer #3 (Remarks to the Author):

The authors have replied my comments and concerns. Now I recommend to publish this work without further review.

Reviewer #4 (Remarks to the Author):

The authors have addressed all concerns and I believe the manuscript in its present form fully deserves publication in this journal.

I would also like to provide an alternative view to Reviewer 1's comments. The manuscript submitted here is a strong demonstration of what can be done with MLIPs and dismissing it as part of a "wave", lacking novelty, and not of high enough interest for the broad audience of Nature Communications I believe does not do it justice.

The work aggregates a massive amount of reference data (a plus if they are shared, not a minus, since the speedup is still large), and trains an interatomic potential on a highly complex, *reactive* system. This has been difficult to do for decades, not only for pre-ML potentials, but still remains a challenge for ML potentials and most of the methods in this "wave" would struggle to do so. This work demonstrates that this is indeed possible and is done in a remarkably thorough way. The way the authors were able to immediately apply the potential the reviewer's suggestion to study the kinetic isotope effect and found no degradation in performance on the 1,000 extracted structures is a strong demonstration of precisely what this potential enables.

Reviewer 1

I am happy with the revised manuscript and now recommend publication. The calculation of isotope effects adds important data to the paper that can in principle be measured.

We would like to thank again the Reviewer for the excellent suggestion of computing the reaction kinetic isotope effect, which has allowed us to strengthen the original message of the paper. We are very happy that the manuscript quality is now considered suited for publication.

Reviewer 2

The revised version of this work has addressed my questions, and I now recommend acceptance.

The wording regarding the comparison between experimental work on real materials and computational modeling on idealized structures has been improved, with a prospect for further theoretical work in the area.

I was still hesitating to recommend the paper, because the authors claim, on the one hand, that this is important for catalysis, but then they do not show models that make use of their methods in catalysis. This is a model study of a proton hopping process between oxygen atoms in an AlO₄ site, and I cannot imagine that this would be of relevance in a catalytic reaction. The two examples the authors mention now are the dehydration of an alcohol and methane oxidation on a Cu site. The alcohol OH group and the water product will certainly impact the motion of the zeolite proton, and methane oxidation is a quite different case, because a C-H bond is broken (and not O-H).

However, the finding that quantum effects do play a role at relevant catalytic temperatures in zeolites, and the methods used here are paving the way for future studies in catalysis are deemed highly important by this reviewer. It is the first study of quantum effects of proton hopping in zeolites, to the best of my knowledge, and the revised version now also has added a study on kinetic isotope effects, making the concept amenable for an experimental comparison.

We thank the reviewer for the thorough evaluation of the revised manuscript. While it is indeed true that proton hopping is not really a catalytic process, we believe that it remains an important model reaction to investigate the propensity of a zeolite framework to transfer its proton to a substrate triggering a Brønsted-acid catalysed reaction. While of course this also depends on the host-guest interactions with the rest of the framework atoms, the proton hopping kinetics can provide insights in the strength of the O-H bond and, therefore, its acidity.

The two examples of zeolite-catalyzed reactions reported in the manuscript were not chosen because they resemble the proton hopping in the pristine framework, as correctly pointed out by the Reviewer. On the other hand, they were chosen because they do involve proton transfer reactions and, therefore, if a computational investigation would be performed NQEs should be taken into account when discussing the kinetic results.

We are very pleased that, despite this remark, the Reviewer still deems the manuscript suitable for publication.

Reviewer 3

The authors have replies my comments and concerns. Now I recommend to publish this work without further review.

We thank the Reviewer again for his/her helpful suggestions and for this final positive evaluation of the manuscript.

Reviewer 4

The authors have addressed all concerns and I believe the manuscript in its present form fully deserves publication in this journal.

I would also like to provide an alternative view to Reviewer 1's comments. The manuscript submitted here is a strong demonstration of what can be done with MLIPs and dismissing it as part of a "wave", lacking novelty, and not of high enough interest for the broad audience of Nature Communications I believe does not do it justice.

The work aggregates a massive amount of reference data (a plus if they are shared, not a minus, since the speedup is still large), and trains an interatomic potential on a highly complex, *reactive* system. This has been difficult to do for decades, not only for pre-ML potentials, but still remains a challenge for ML potentials and most of the methods in this "wave" would struggle to do so. This work demonstrates that this is indeed possible and is done in a remarkably thorough way. The way the authors were able to immediately apply the potential the reviewer's suggestion to study the kinetic isotope effect and found no degradation in performance on the 1,000 extracted structures is a strong demonstration of precisely what this potential enables.

We would like to thank the Reviewer for the kind words and the appreciation for the work that has been put in this manuscript. We are glad that the new simulations were able to satisfactorily reply to the concerns of Reviewer 1, which has now given positive advice for the manuscript publication.